



# Aptian-Albian clumped isotopes from northwest China: Cool temperatures, variable atmospheric $p$CO$_2$ and regional shifts in hydrologic cycle

Dustin T. Harper[1], Marina B. Suarez[1,2], Jessica Uglesich[2], Hailu You[3,4,5], Daqing Li[6], Peter Dodson[7]

[1] Department of Geology, The University of Kansas, Lawrence, KS, U.S.A.
[2] Department of Geological Sciences, University of Texas, San Antonio, TX, U.S.A.
[3] Key Laboratory of Vertebrate Evolution and Human Origins, Institute of Vertebrate Paleontology and Paleoanthropology, Chinese Academy of Sciences, Beijing, P.R.C.
[4] Chinese Academy of Science Center for Excellence in Life and Paleoenvironment, Beijing, P.R.C.
[5] College of Earth and Planetary Sciences, University of Chinese Academy of Sciences, Beijing, P.R.C.
[6] Institute of Vertebrate Paleontology and College of Life Science and Technology, Gansu Agricultural University, Lanzhou, P.R.C.
[7] Department of Biomedical Sciences, The University of Pennsylvania, Philadelphia, U.S.A.

*Correspondence to*: Dustin T. Harper (dtharper@ku.edu)

**Abstract.** The Early Cretaceous is characterized by warm background temperatures (i.e., greenhouse climate) and carbon cycle perturbations that are often marked by Ocean Anoxic Events (OAEs) and associated shifts in the hydrologic cycle. Higher-resolution records of terrestrial and marine $\delta^{13}$C and $\delta^{18}$O (both carbonates and organics) suggest climate shifts during the Aptian-Albian, including a warm period associated with OAE 1a in the early Aptian and subsequent "cold snap" near the Aptian-Albian boundary prior to the Kilian and OAE 1b. Understanding the continental system is an important factor in determining the triggers and feedbacks to these events. Here, we present new paleosol carbonate stable isotopic ($\delta^{13}$C, $\delta^{18}$O and $\Delta_{47}$) and CALMAG weathering parameter results from the Xiagou and Zhonggou Formations (part of the Xinminpu Group in the Yujingzi Basin of NW China) spanning the Aptian-Albian. Published mean annual air temperature (MAAT) records of the Barremian-Albian from Asia are relatively cool with respect to the Early Cretaceous. However, these records are largely based on coupled $\delta^{18}$O measurements of dinosaur apatite phosphate ($\delta^{18}$O$_p$) and carbonate ($\delta^{18}$O$_{carb}$), and therefore rely on estimates of meteoric water $\delta^{18}$O ($\delta^{18}$O$_{mw}$) from $\delta^{18}$O$_p$. Significant shifts in the hydrologic cycle likely influenced $\delta^{18}$O$_{mw}$ in the region, complicating these MAAT estimates. Thus, temperature records independent of $\delta^{18}$O$_{mw}$ (e.g., clumped isotopes or $\Delta_{47}$) are desirable, and required to confirm temperatures estimated with $\delta^{18}$O$_p$ and $\delta^{18}$O$_c$, and to reliably determine regional shifts in $\delta^{18}$O$_{mw}$. Primary carbonate material was identified using traditional petrography, cathodoluminescence inspection, and $\delta^{13}$C and $\delta^{18}$O subsampling. Our preliminary $\Delta_{47}$-based temperature reconstructions (record mean of 14.9 ºC), which we interpret as likely being representative of MAAT, match prior estimates from similar paleolatitudes of Asian MAAT (average ~15 ºC) across the Aptian-Albian. This, supported by our estimated mean atmospheric paleo-$p$CO$_2$ concentration of 396 ppmv, indicates relatively cooler mid-latitude terrestrial climate. Additionally,



our coupled $\delta^{18}O$ and $\Delta_{47}$ records suggest shifts in the regional hydrologic cycle (i.e., $\Delta MAP$ and $\Delta\delta^{18}O_{mw}$) that may track Aptian-Albian climate perturbations (i.e., a drying of Asian continental climate associated with the cool interval).

## 1 Introduction

Early Cretaceous climate is characterized by a warm background greenhouse climate state and perturbations to climate and the carbon cycle associated with shifts in global $\delta^{13}C$, including Cretaceous Ocean Anoxic Events (OAEs; Föllmi, 2012;
Hay, 2016; Jenkyns, 2018). Such climate aberrations can provide insight into the sensitivities and coupling of the carbon cycle, climate and the hydrologic cycle through quantitative reconstruction of past physical and environmental conditions (e.g., atmospheric paleo-$p$CO$_2$, temperature and precipitation). Indeed, much work has been done generating geochemical proxy-based observations and simulations of past global warming events which serve as useful analogues of future warming (e.g., Hönisch et al., 2012; Zachos et al., 2008; Hay, 2016). Similarly, both long-term and abrupt cooling intervals in the past
can supply proxy-based observations of negative climate feedbacks associated with carbon sequestration and global cooling.

Multiple climate events (including OAEs) have been identified during the late Early Cretaceous (Vickers et al., 2019; Jenkyns, 2018) oftentimes referred to as the mid-Cretaceous (i.e., here, our records span the Aptian-Albian; 125 to 100.5 Ma). While limited, available temperature records (e.g., Jenkyns, 2018) and high latitude sedimentological evidence (Vickers
et al., 2019) suggest a relatively cool interval (following warmth during OAE 1a) associated with a global carbon isotope maximum (i.e., "C10"; Bralower et al., 1999; Menegatti et al., 1998; Mutterlose et al., 2009) at the Aptian-Albian boundary prior to OAE 1b (Bottini et al., 2015). Estimates of Aptian-Albian atmospheric paleo-$p$CO$_2$, while highly uncertain, tend to suggest low (less than 1000 to 1500 ppmv background greenhouse climate conditions; Franks et al., 2014) concentrations at the Aptian-Albian consistent with a cooler climate (Du et al., 2018; Passalia, 2009; Haworth et al., 2010; Aucour et al., 2008;
Ekart et al., 1999; Wallmann, 2001; Fletcher et al., 2005). This C10 interval has been identified on land using stable isotopes in terrestrial paleosol carbonates and organic carbon from the continental interiors of North America (Suarez et al., 2014; Ludvigson et al., 2010) and Asia (Suarez et al., 2018). For Asia, Aptian-Albian terrestrial temperature estimates have been generated using oxygen isotopes in dinosaur tooth enamel (Amiot et al., 2011) and suggest a relatively cool interval (e.g., ~10 ± 4 ºC mean annual air temperature, MAAT; ~42 ºN paleolatitude) consistent with marine paleotemperatures
(Mutterlose et al., 2009; Bottini et al., 2015) and elevated global $\delta^{13}C$ (Menegatti et al., 1998; Bralower et al., 1999). However, MAAT estimates from $\delta^{18}O$ of dinosaur tooth enamel phosphate ($\delta^{18}O_p$) hinge on the relationship between mean annual temperature, latitude and the $\delta^{18}O$ of meteoric water or $\delta^{18}O_{mw}$ (Amiot et al., 2004). $\delta^{18}O_{mw}$ is influenced by other parameters in addition to temperature and latitude, and is further complicated as the intensity of poleward moisture transport is altered by greenhouse climate conditions. Therefore, confirming these temperatures with a secondary geochemical proxy
is warranted.



Hydrologic cycle models and observations of past warm intervals (e.g., early Cenozoic and greenhouse Cretaceous) indicate an "intensification" of the hydrologic cycle due to enhanced poleward moisture transport associated with global warming (e.g., Carmichael et al., 2015; Hasegawa et al., 2012; Suarez et al., 2011a; White et al., 2001; Poulsen et al., 2007). Likewise,

as temperatures cool during Cretaceous climate recovery or during long-term transitions driven by changes in global tectonics and paleogeography, the hydrologic cycle tends to respond with regionally dependent shifts in mean annual precipitation (MAP). For example, Hasegawa et al. (2012) observed hydrologic cycle responses track greenhouse gas (GHG) forcing in Asia during Aptian-Albian. For the Aptian-Albian, models and observations suggest changes in continental interior precipitation during the global "cold snap" (e.g., Mutterlose et al., 2009) and the potential for variable Asian aridity

associated with warm/cool cycles (Hasegawa et al., 2010; Hasegawa et al., 2012; Föllmi, 2012; Poulsen et al., 2007; Tabor et al., 2016; Zhou et al., 2008), which may hamper $\delta^{18}O_p$-based temperature reconstructions for the Aptian-Albian that fail to quantify $\delta^{18}O_{mw}$ independently of $\delta^{18}O_p$. To address this deficiency, here we provide new multi-proxy records from the Yujingzi Basin of NW China spanning the Aptian-Albian using $\delta^{13}C$, $\delta^{18}O$ and $\Delta_{47}$ (i.e., clumped isotopes) of terrestrial paleosol carbonates. Additionally, MAP is quantified using chemical weathering ratios, specifically CALMAG (Nordt and

Driese, 2010). We combine our new records with organic stable carbon isotope chemostratigraphic records for the site (Suarez et al., 2018) to provide age control to quantitatively interpret shifts in regional temperature, $\delta^{18}O_{mw}$, MAP, and global atmospheric paleo-$p$CO$_2$ associated with the Aptian-Albian. These proxy interpretations are compared to models and records of Cretaceous Asian climate and the global exogenic carbon cycle (i.e., atmospheric paleo-$p$CO$_2$) to provide new constraints on Aptian-Albian climate, carbon and hydrologic cycles.

## 2 Materials and methods

### 2.1 Sampling and analyses

The Xiagou and Zhonggou Formations, part of the Xinminpu Group in the Yujingzi Basin of Northwest China (Gansu Province), were sampled in 2011 with the goal of placing the Early Cretaceous paleobiology and geology of this region in a global climate and carbon isotope chemostratigraphic framework (e.g., Suarez et al., 2018). The Xinminpu Group,

approximately Early Cretaceous in northwest China, is composed of four formations (ordered stratigraphically bottom to top): Chijinqiao Formation, Chijinpu Formation, Xiagou Formation, and Zhonggou Formation. Outside of the Yujingzi Basin, Xinminpu group strata produce Aptian radiometric dates of 123.0 ± 2.6 to 133.7 ± 1.8 Ma (Li et al., 2013; Kuang et al., 2013). Outcrop sections (Fig. 1) are regionally exposed in the Yujingzi basin at a fossil-rich site informally known as the White Pagoda Site, produced by accommodation of strike-slip motion from Lhasa Block convergence with Asia (Chen and

Yang, 1996; Vincent and Allen, 1999).

Outcrop sections for White Pagoda were numbered and split into three facies by Suarez et al. (2018): 1) the lowermost facies consisting of sections 1, 2, 2A, 2) an overlying facies of alternating gray and variegated mudstones and muddy sandstones





consisting of sections 3, 3A-H, and 6, and 3) coarse-grained poorly sorted arkosic sandstones and sandy mudstones (Section
4). Here, we utilize sample material from sections (in stratigraphic order) 3, 3A-H, 6, and 4. Suarez et al. (2018) observed
carbonate nodules, root-traces, charophytes, turtle remains, ostracodes and gastropods within the middle facies (Sections 3,
3A-H, and 6), with root-traces and nodules extending into the uppermost facies (Section 4). Section 4 had a higher degree of
color-mottling, blocky ped-structures and burrows compared to the underlying facies. Facies interpretations for sections
sampled for this study indicate fluvio-lacustrine and palustrine environments (i.e., Suarez et al., 2018). For example, sections
3, 3A-H, 6, and 4 exhibit evidence of subaerial exposure (soils indicated by horizonation, slickensides, root traces and
carbonate nodule formation), fluvial deposition (lenticular sands fining up with erosive bases), and lacustrine environments
(turtle remains, charophytes, ostracodes, thin limestones, and organic-rich mudstones) (Suarez et al., 2018). For the sections
of interest, the presence of cracking, slickensides (mukkara structures) and expansive clays suggests wet/dry periods typical
of vertisols.


The organic stable carbon isotope record at White Pagoda was used as a guide to sample carbonate nodules for analysis, at 8
to 60 meter intervals, consistently with global Aptian-Albian carbon isotope chemostratigraphy (i.e., samples from intervals
in which regional trends in organic $\delta^{13}$C were consistent with higher-resolution records of Bralower et al. (1999) and
Menegatti et al. (1998)). Nodules for analysis were sampled from well below channel sands and surface paleosol horizons in
outcrop (i.e., sampled from paleosol B horizons; Tabor and Myers, 2015) to help avoid potential surficial biases on carbonate
(e.g., radiative heating in soil depths <50 cm; Burgener et al., 2019). Paleosols from which nodules are sampled are fine
grained throughout the section which suggests suitability for clumped isotope-based MAAT interpretation (e.g., Kelson et
al., 2020).

Thin sections were cut from hand samples for petrographic analysis and cathodoluminescence (CL) imaging to aid in the
identification and isolation of primary carbonate nodule material. Briefly, thin sections were inspected for environmental
indicators and microfabrics, and photographed under plane-polarized and cross-polarized light (PPL and XPL, respectively)
using an Olympus BX43P petrographic microscope with a SC50 Olympus camera. Thin sections were then CL imaged using
a Relion Industries Reliotron III cold-cathode chamber, with operating conditions consisting of a rarified helium atmosphere
at 50 milliTorr, accelerating voltage of 10 kV, and beam current of 0.5 mA. Macroscale imaging through the 50 mm top
window of the chamber was carried out using a 16 Mpx Canon EOS SL1 DSLR camera with a macro lens suspended over
the CL chamber. CL imaging was used to detect any heterogeneities in cation substitution which may indicate alteration, as
$Mn^{2+}$ tends to substitute for $Ca^{2+}$ in reducing conditions generating bright orange luminescence (Habermann et al., 2000;
Cazenave et al., 2003).


Once located in thin section using petrography and CL, primary nodule carbonate was mapped onto the corresponding thin-
section billet, and sampled using a dental drill. In some samples, suspect non-primary carbonate material (e.g., spar) was also



sampled for isotopic comparison, but excluded from primary carbonate isotopic averages reported here. Depending on the number of nodules and lithologic complexity of a hand sample, approximately 8 to 14 ~50 μg samples were drilled for
traditional stable isotopes (i.e., $\delta^{13}C$ and $\delta^{18}O$) in each hand sample. Stable isotope samples were heated to 200 ºC in a vacuum for 1 hour prior to analysis via ThermoFinnigan MAT 253 gas source isotope ratio mass spectrometer (IR-MS) coupled to a Kiel IV carbonate device at the University of Kansas (KU) Keck-NSF Paleoenvironmental and Environmental Stable Isotope Laboratory (KPESIL). Standard reproducibility indicates analytical precision (1σ) of 0.03‰ and 0.05‰ for $\delta^{13}C$ and $\delta^{18}O$, respectively.


Following $\delta^{13}C$ and $\delta^{18}O$ analysis of dental-drilled carbonate powder, larger samples (~6 mg of material for each analysis; n = 3 to 4 per sample), were drilled for clumped isotope ($\Delta_{47}$) analysis from areas of primary nodule carbonate exhibiting uniform $\delta^{13}C$ and $\delta^{18}O$, and CL. Clumped isotopes were measured at the University of Colorado Boulder (UCB) Earth Systems Stable Isotope Laboratory (CUBES-SIL) on a custom automated vacuum line sample introduction system, in which
samples are digested at 90 ºC in a common phosphoric acid bath. This system removes isobaric contamination by entraining the $CO_2$ sample in helium and passing it through a ~1.5 m long stainless steel column hand packed with Poropak for 45 minutes at −20 °C. $CO_2$ is transferred to the sample side bellow of a ThermoFinnigan dual inlet MAT 253+. Values are reported relative to the carbon dioxide equilibration scale (Dennis et al., 2011), using gases with a range of bulk $\delta^{47}$ values and equilibrated at 1000 °C and 25 °C to convert in-house values to the CDES scale. IUPAC parameters for $^{17}O$ corrections
(Brand et al., 2010) were used in the initial steps of data reduction, following recommendations of (Daëron et al., 2016) and (Schauer et al., 2016). We then applied an acid correction factor (0.088‰) appropriate for use with values calculated using IUPAC parameters (Petersen et al., 2019). International standards (i.e., ETH1, ETH2, ETH3, ETH4, IAEA-C1, IAEA-C2, Merck and NBS19) were utilized to further correct $\Delta_{47}$ values. Potentially contaminated data was culled (e.g., sample analyses which exhibit $\Delta_{48}$ excess that tracks variability in $\Delta_{47}$; see Supplementary Material).


X-Ray Fluorescence (XRF) measurements were carried out on samples from horizons that appear to be well-developed paleosols, specifically horizons interpreted as B-horizons. Analysis was completed with a Rigaku Primus II WD-XRF spectrometer at the University of Texas at San Antonio. Raw X-ray intensities were calibrated by the analysis of eight USGS certified elemental standards (BIR-1a, COQ-1, DNC-1a, GSP-2, RGM-2, SBC-1, STM-2, W-2a), with an RSD value of
0.036%. Weight percentages were converted into molar weights before application of a chemical index, following Sheldon and Tabor (2009). $Al_2O_3$, CaO, and MgO are the oxides used for calculation of the CALMAG (Nordt and Driese, 2010) chemical weathering index (see following section for parameter calculation and proxy details).

### 2.2 Quantitative proxies

Clumped isotopes (i.e., $\Delta_{47}$) have been successfully utilized to estimate temperature in carbonates, leveraging the
thermodynamically controlled abundance of isotopically heavy $^{13}C$ and $^{18}O$ bonded isotopes (Ghosh et al., 2006; Schauble et





al., 2006). This approach has an advantage over $\delta^{18}$O-based temperature estimates, as other controlling variables (e.g., $\delta^{18}O_{mw}$) need not be estimated. $\Delta_{47}$ values are translated into calcification temperature following the calibration of Petersen et al. (2019) and we define our temperature uncertainty as $1\sigma$ of replicate analyses. Additional temperature calibration approaches (i.e., Ghosh et al., 2006; Bonifacie et al., 2017) and calculation details (i.e., R code for data analysis) are

available in the Supplementary Material. However, for this study, in subsequent calculations and figures, we opt for Petersen et al. (2019) $\Delta_{47}$ values and calibration temperatures calculated using the following relationship:

$$\Delta_{47} = (0.0383 \pm 1.7^{-6}) \times (10^6 / T^2) + (0.258 \pm 1.7^{-5}) \tag{1}$$

Groundwater $\delta^{18}$O is derived from the oxygen isotopic composition of precipitation which is ultimately controlled by factors

such as temperature, amount, continentality and seasonality. It can be further modified by processes such as evaporation in paleoenvironments which experience wet/dry cycles. $\delta^{18}$O of groundwater ($\delta^{18}O_w$) can be determined for pedogenic carbonate calcification once temperature is known and $\delta^{18}O_{carb}$ is measured following Friedman and O'Neil (1977):

$$\delta^{18}O_w \text{ (SMOW)} = (\delta^{18}O_{carb} \text{ (SMOW)} + 10^3) / (e^{(18030 / T - 32.42) / 1000}) - 10^3 \tag{2}$$

To estimate mean regional precipitation for the study interval and determine shorter-term precipitation variability in our record, we use the bulk geochemical compositional proxy CALMAG (Nordt and Driese, 2010), which utilizes the gains and losses of elemental oxide abundances as a result of weathering in vertisols. The concentration of aluminum oxide, calcium oxide and magnesium oxide are estimated using XRF and the CALMAG parameter is determined:

$$\text{CALMAG} = (Al_2O_3) / (Al_2O_3 + CaO + MgO) \times 10^2 \tag{3}$$

Mean annual precipitation (MAP) is then determined from the CALMAG parameter based on the Nordt and Driese (2010) calibration:

$$\text{MAP} = 22.69 \times (\text{CALMAG}) - 435.8 \tag{4}$$

Paleosols have been widely utilized as archives to determine the past concentration of atmospheric $p$CO$_2$ (Ekart et al., 1999;

Cerling, 1991). While requiring a number of assumptions, soil carbonate nodule $\delta^{13}$C, when used in tandem with estimates from other proxies (e.g., MAP from CALMAG and respired soil $\delta^{13}C_{CO2}$ from $\delta^{13}C_{org}$), provide many of the most robust estimates of Cretaceous atmospheric $p$CO$_2$ outside of a stomatal approach (Franks et al., 2014), especially because paleosol carbonate nodules are abundant in the rock record. The soil carbonate paleobarometer uses a diffusion model in which atmospheric $p$CO$_2$ ($\delta^{13}C_a$) and respired CO$_2$ from soils ($\delta^{13}C_r$) are the dominant controls on soil CO$_2$ ($\delta^{13}C_s$) following the

mixing model of Cerling (1991) in terms of $\delta^{13}$C (Ekart et al., 1999). The relative isotopic influence of atmospheric versus respired soil CO$_2$ on soil CO$_2$ (i.e., the source CO$_2$ for calcite) will therefore be controlled by the concentration of CO$_2$ in the atmosphere, if the concentration of the soil-derived component of total gas at depth, S(z), is accounted for following Ekart et al. (1999):

$$p\text{CO}_2 = S(z) \times ((\delta^{13}C_s - 1.0044 \times \delta^{13}C_r - 4.4 / (\delta^{13}C_a - \delta^{13}C_s)) \tag{5}$$





$\delta^{13}C_s$ can be determined from $\delta^{13}C_{carb}$, assuming temperature-dependent fractionation (here we use $\Delta_{47}$-based temperature) between gaseous soil $CO_2$ and carbonate (Romanek et al., 1992). Suarez et al. (2018) correlated sections in this study to bulk carbonate surface marine sections using $\delta^{13}C$ of organic carbon. We estimate atmospheric $\delta^{13}C$ (i.e., $\delta^{13}C_a$) from a marine section correlated chemostratigraphically with the White Pagoda Site (i.e., Peregrina Canyon, Mexico of Bralower et al. (1999) correlated to White Pagoda by Suarez et al. (2018)), applying a $\delta^{13}C_{DIC}$ (i.e., $\delta^{13}C$ of marine dissolved inorganic

carbon, DIC) to $\delta^{13}C_a$ fractionation of –8.23‰ consistent with "greenhouse climate" carbon cycle simulations (i.e., Zeebe, 2012), and assuming bulk carbonate $\delta^{13}C$ for the Peregrina Canyon section is representative of global surface DIC $\delta^{13}C$. For $\delta^{13}C_r$, we apply the bulk sedimentary organic carbon $\delta^{13}C$ values of Suarez et al. (2018).

In addition to estimates of $\delta^{13}C$ for the three carbon reservoirs outlined above, the term S(z), or the depth-dependent

contribution of soil-respired $CO_2$, must be determined to compute atmospheric paleo-$pCO_2$. While this term is a significant source of uncertainty due in part to a large range of potential past environmental conditions, Cotton and Sheldon (2012) hypothesized a relationship between summer minimum S(z) and MAP using observations of modern soils:

$$S(z) = 5.67 \times MAP - 269.9 \tag{6}$$

Here, we apply their relationship to compute S(z) from our CALMAG-based MAP estimates. It is important to note that the

relationship defined by Cotton and Sheldon (2012) uses a dataset which does not include humid climate soils or vertisols, and it is therefore cautiously applied and discussed in terms of paleoenvironmental influence on our paleo-$pCO_2$ estimates (i.e., we evaluate our atmospheric $pCO_2$ record against a large range in S(z)).

## 3 Results

### 3.1 Petrography

Based on carbonate petrography we recognize two distinct microfacies in our samples and split samples into two groups (microfacies (i) and (ii)) to evaluate the origin of stable isotope values (primary vs. secondary; depositional environment) (Fig. 2; Table 1). Microfacies (i) is characterized by distinct nodules which originated from primarily clayey horizons, consisting of dense micrite with abundant root traces and fractures filled with sparry calcite and microspar calcite (Fig. 2). Fracturing is less pervasive and micritic nodules include less clay minerals in microfacies (i). Nodule micrite is brightly

luminescent under CL. The clay matrix displays birefringent microfabric and contains subangular to subrounded clasts of calcic and siliciclastic grains (mainly quartz and feldspars as well as fragments of other nodules) (Fig. 3). Microfacies (i) includes samples 3-021, 3A-097, 4-038, and 3H-014.

Samples in microfacies (ii) (observed in samples 3B-021, 3E-001, 6-003, and 6-042), tend to be coalesced nodules or beds

comprised almost entirely of clay-rich microcrystalline calcite in which discreet nodules are less evident (e.g., samples 3E-001 and 6-042; Fig. 3). The micrite that comprises carbonate nodules in microfacies (ii) tends to be duller with regard to CL



luminescence (Fig. 3).The second microfacies is characterized by clay-rich (i.e., common to frequent in abundance) micritic limestone with abundant fracturing and brecciation, including circum-granular fractures (i.e., sample 6-042; Fig. 3). These are filled with microspar and spar (Fig. 2). Color mottling and Mn-staining are observed, perhaps related to microbial

activity (i.e., thrombolites and/or pisoids; Fig. 2b). Sample 3F-019 appears to be a mixture of the two microfacies, with dense, brightly luminescent (i.e., CL) micritic nodules in dominantly clay matrix, mottled coloring in thin section, rhizoliths, circum-granular fractures and Mn-staining (Fig. 2 panel c). Nodules appear slightly coalesced in this sample (i.e., unlike microfacies (i)), whilst individual nodule shape is somewhat maintained (i.e., unlike microfacies (ii)).

### 3.2 Traditional stable and clumped isotopes

Stable isotopes of drill spot samples show a high degree of intrasample homogeneity (Fig. 4). Measurements between University of Kansas and University of Colorado, Boulder are largely consistent with comparable precision (Tables 1 and 2; Fig. 4). $\delta^{13}C$ values range from −8‰ to −3‰ and $\delta^{18}O$ ranges from −12‰ to −6‰ for carbonates measured in this study. Sample 3B-021 displays the most heavy-isotope enriched $\delta^{13}C$ and $\delta^{18}O$ values, with $\delta^{13}C$ more than 2‰ and $\delta^{18}O$ more than 1‰ greater than all other samples (Fig. 4), despite the relative isotopic low in the $\delta^{13}C_{org}$ curve which results in a large $\Delta^{13}C$

for that sample (Table 1). Carbonate samples tend to be isotopically homogenous ($2\sigma \leq 0.6‰$ for all sample $\delta^{13}C$ and $\delta^{18}O$, with only 2 samples with $2\sigma > 0.3‰$; Table 1) following Cotton and Sheldon (2012), who proposed a requirement of $2\sigma < 0.5‰$ for $\delta^{13}C$ and $\delta^{18}O$ for all samples applied to paleo-$p$CO$_2$ reconstructions. We discern no relationship between $\delta^{13}C$ and $\delta^{18}O$ of carbonates, nor grouping of microfacies by stable isotopic composition (e.g., Fig. 4; Table 1).

Clumped isotope ($\Delta_{47}$) mean sample values range from 0.707 to 0.732 (Table 2) which, following the Petersen et al. (2019) calibration, translates to temperatures ranging from ~10 to 20 ℃, with an average temperature of 14.9 ℃ for the entire record. Transient cooling of ~2 to 4 ℃ (i.e., down to 11.1 ℃) is observed in the C10 carbon isotope interval, with the warmest temperature occurring immediately following the C10 interval (i.e., warms to 18.8 ℃; Fig. 5).

### 3.3 CALMAG

CALMAG values for all measured samples range from a low of 2% to a high of 70%. Lowest values are either samples that were not identified as B-horizons or likely immature soils which yield values inapplicable to range in calibration (CALMAG less than ~35%; Table S4). If only B-horizon samples applicable to the range in the Nordt and Driese (2010) calibration are considered, maximum variability in CALMAG is ± 12% (Table 3; Table S4). This translates to MAP variability of ± 270 mm/yr over the interval, with mean MAP of 641 mm/yr (i.e., mean CALMAG of 47.5%) for paleosols in which clumped

isotopes were also measured (Table 3; Fig. 5).



## 4 Discussion

### 4.1 Carbonate nodule δ¹³C and δ¹⁸O

Light stable isotopes ($\delta^{13}C$ and $\delta^{18}O$) of carbonate material measured at KU and CUB are consistent (Tables 1 and 2; Fig. 4) indicating primary carbonate was successfully sampled from nodules for clumped isotope analyses (i.e., primary carbonate

isotopic composition characterized by drill spot measurements at KU match values from CUB clumped measurements). $\delta^{13}C$ in carbonate nodules is controlled by soil water DIC which, through time, is ultimately controlled by variation in the other exogenic carbon reservoirs. Carbonate $\delta^{18}O$ is reflective of regional meteoric water and temperature. Though much coarser resolution, our carbonate $\delta^{13}C$ largely follows $\delta^{13}C_{org}$ which has been tied to global variations in the carbon cycle (Suarez et al., 2018; Ludvigson et al., 2010; Ludvigson et al., 2015; Heimhoffer et al., 2003; Ando et al., 2002), suggesting both

carbonate and organic records at the site track global variability in the carbon cycle originally described in Menegatti et al. (1998) and Bralower et al. (1999) (Fig. 5) (e.g., $\delta^{13}C_{carb}$ is highest in the C10 interval). We observe no clear grouping of carbonate stable isotopes by microfacies and all samples contain pedogenic features. This suggests $\delta^{13}C_{carb}$ tracks global variations in the carbon cycle, and $\delta^{18}O_{carb}$ values are reflecting $\delta^{18}O$ of regional precipitation once temperature is considered.

### 4.2 Interpreting paleoenvironmental biases in Δ₄₇-based temperatures

Macroscopic features described in Suarez et al. (2018) along with traditional carbonate petrography suggest a paleoenvironment which experienced wet/dry cycles. These features include redoximorphic color mottling, gilgai structures, fracturing pervasive to varying degrees in carbonate nodules, microspar and spar recrystallization present in voids/fractures, Mn staining and root traces (Fig. 2 and 3). Microscopic features are consistent with facies interpretations of Suarez et al.

(2018) which suggest fluvio-palustrine paleoenvironment. Rhizoliths (i.e., calcified root structures) in nearly all nodule samples (e.g., Fig. 2) indicate that vegetation was present and the carbonate nodules are indeed soil-formed in subhumid to semiarid conditions (Zhou and Chafetz, 2009). Indeed, mean MAP derived from our CALMAG proxy record suggests 712 mm/yr (respective minimum and maximum MAP of 476 and 984 mm/yr for the interval; Fig. 5) and $\Delta_{47}$-based temperatures range from 11.4 ± 4.8 °C to 18.8 ± 2.2 °C, consistent with the subhumid to semiarid environments in which soil carbonates

commonly form (Zhou and Chafetz, 2009; Birkeland et al., 1999; Breecker et al., 2009).

Understanding the timing of carbonate formation in soils is important for interpretation of $\delta^{13}C$, $\delta^{18}O$ and $\Delta_{47}$. The solubility of calcite is the primary controlling factor on carbonate formation, and it is significantly affected by soil $CO_2$ concentration. Because $CO_2$ concentration is lower in warmer conditions, and drier conditions result in greater concentration of ions, calcite

precipitation tends to occur during warm, dry conditions. Numerous early studies have suggested warm season bias in soil carbonate formation and thus the $\Delta_{47}$-derived temperatures (Breecker et al., 2009; Passey et al., 2010). Recent work of Kelson et al. (2020) suggests this may not always be the case for a number of reasons. The presence of vegetation (suggested



by abundant root traces) may shade the soil surface from solar radiation. However, Burgener et al. (2019) and Kelson et al. (2020) found that this effect is rare, and samples for this study were collected from paleosol horizons deep enough (i.e., > 50

cm) to be buffered against the effects of radiative heating (i.e., Burgener et al., 2019). Seasonality of precipitation, evaporation, and evapotranspiration likely affects the degree to which a warm season temperature bias may occur. In a study of modern soils in North America, Gallagher and Sheldon (2016) suggested that only continental climate with rainy seasons in the spring had summer temperature biases. Suarez et al. (2011b) suggested that lower than expected temperatures of Mio-Pliocene soil carbonates from the Chinese Loess Plateau may be the result of a monsoon climate in which the rainy seasons

occur during the warmest part of the season and conditions for calcite precipitation occurs prior to or after the warm season.

These studies suggest that carbonate nodule clumped isotope-based temperatures revealed from the Xinminpu Group likely represent lower temperatures than mean warm season. In addition, mean clumped isotope-based temperature over the study interval (14.9 ºC) matches Aptian-Albian MAATs derived from phosphate $\delta^{18}O$ in dinosaur teeth from similar paleolatitudes

in Asia (i.e., 15 ºC for Xinminpu group; Amiot et al., 2011). However, our mid-latitude continental interior temperatures reflect the temperature of calcite precipitation and may be biased towards the time of year during which a region experiences its first month without water storage, which varies by regional climate (Gallagher and Sheldon, 2016). Given our paleoenvironmental interpretation of wet-dry seasonality which resulted in vertisol formation at our study location, and proxy-based estimates of MAAT and MAP, the paleoenvironment is likely best-represented by either the "continental" or

"semi-arid monsoonal" climates of Gallagher and Sheldon (2016). We note that the modern soil type for the settings of Gallagher and Sheldon (2016) consists of mollisols and thus may not be representative of the vertisols in which nodules used in this study formed. Their "continental" model indicates a decline in water storage in July/August which tends to bias carbonate formation to warmer values. In contrast, the "semi-arid monsoonal" model shows a decrease in water storage in April resulting in a slight cool season bias. However, cool season biases tend to be much smaller in magnitude (less than 4

ºC) than warm season biases (as much as 24 ºC) (Kelson et al., 2020). Therefore, regardless of the interpretation of seasonal biases, our mean temperature based on clumped isotopes (14.9 ºC) suggests very cool conditions in the mid-latitude Asian continental interior during the Aptian-Albian. Any potential warm season bias on our temperature results is unlikely as it would suggest even cooler conditions inconsistent with combined proxy observations. Indeed, the temperatures calculated here are consistent with other regional paleotemperature proxy observations (e.g., Amiot et al., 2011), and counter to the

predominantly warm greenhouse climate of the Cretaceous (Föllmi, 2012).

### 4.3 Latitudinal gradients of temperature and $\delta^{18}O_{mw}$ for the Aptian-Albian

Clumped isotope-based temperatures for the White Pagoda site indicate a mean record value of 14.9 ºC which is equivalent to $\delta^{18}O_p$-based temperature estimates (15 ºC) carried out on dinosaur teeth from formations within the same group (Xinminpu) in NW Asia (Amiot et al., 2011). The groundwater $\delta^{18}O$ based on our combined clumped isotope and carbonate





isotope analyses range from −11.54 to −6.69‰ (VSMOW) and average −9.47, which is somewhat lower than the values of
Amiot et al. (2011) (−7.0‰).

Modern climate observations of the study site indicate cool, dry conditions with mean $\delta^{18}O_{mw}$ of −9.31‰ and −7.66‰ in
nearby Zhangye and Lanzhou, respectively (IAEA/WMO, 2020). Largely due to the influence of regional topography (study

location elevation: ~1500 m), present day precipitation averages < 100 mm/yr and MATs indicate locally cooler
temperatures (9.0 ℃ and 10.5 ℃ in Zhangye and Lanzhou, respectively; IAEA/WMO, 2020) relative to global zonal
averages (15.0 ℃; Rozanski et al., 1993). Aptian-Albian temperatures may similarly be influenced by regional
paleotopography, though topographic reconstructions for Asia during the Aptian-Albian are lacking, limiting speculation.

Generally, proxy-based temperatures and $\delta^{18}O_{mw}$ for the Xinminpu Group tend to fall within zonally averaged general
circulation GENESIS-MOM model results (Zhou et al., 2008) given the large range in possible site paleolatitude during the
Aptian-Albian. For example, paleogeographic reconstructions indicate paleolatitudes ranging from ~35ºN to ~48ºN for the
White Pagoda Site during the Aptian-Albian (Lin et al., 2003; Matthews et al., 2016; Torsvik et al., 2012), which
corresponds to simulated temperatures ranging from 9 to 19 ℃ and simulated $\delta^{18}O_{mw}$ ranging from −11.8 to −6.7‰ (Zhou et

al., 2008).

Combining our new temperature and $\delta^{18}O_{mw}$ data with that compiled in Amiot et al. (2011), we re-cast latitudinal
temperature and $\delta^{18}O_{mw}$ gradients according to the paleogeography applied in Amiot et al. (2011) (Lin et al., 2003) and using
an updated paleogeography based on Matthews et al. (2016) (i.e., Gplates). The updated paleogeography results in higher

Early Cretaceous paleolatitudes for all Asian sites included in this compilation (Supplementary Material; Table S1),
including a more than +13ºN shift for the Xinminpu group sites (Fig. 6). When placed on the paleogeography of Lin et al.
(2003), proxy-based temperature reconstructions for Asia indicate a cool climate relative to latitudinal models of temperature
and hydrology (i.e., land surface gradients compiled in Suarez et al. (2011a) including: leaf physiognomy-based gradients of
Spicer and Corfield (1992), cool and warm Cretaceous gradients of Barron (1983), and GENESIS-MOM general circulation

model gradients of Zhou et al. (2008)). For example, temperature data falls below even the coolest Cretaceous modeled
gradient (i.e., Barron, 1983) despite agreement between proxy $\delta^{18}O_{mw}$ in mid-latitude continental Asia and the modeled cool
Cretaceous (Fig. 7 panels a and b). However, if Matthews et al. (2016) paleolatitudes are applied, proxy-based temperatures
become better aligned with Cretaceous modeled temperature gradients (Fig. 7 panel c). Additionally, the updated
paleolatitudes tend to offset $\delta^{18}O_p$-based $\delta^{18}O_{mw}$ estimates in a positive direction relative to the modeled cool Cretaceous

$\delta^{18}O_{mw}$ gradient, aligning these data with a flatter, more modern appearing $\delta^{18}O_{mw}$ gradient (Fig. 7 panel d). Meanwhile, our
clumped isotope and $\delta^{18}O_{carb}$-based $\delta^{18}O_{mw}$ value falls within error of the range in Cretaceous modeled $\delta^{18}O_{mw}$ gradients (Fig.
7 panel d) suggesting potential errors in required assumptions for $\delta^{18}O_p$-based $\delta^{18}O_{mw}$ reconstructions of Amiot et al. (2011).
The $\delta^{18}O_p$ may be more [18]O-enriched compared to $\delta^{18}O_{mw}$ than accounted for in those original studies. The consumption of





evaporatively-enriched leaf water in herbivores provides one possible mechanism for $^{18}O_P$-enrichment (Levin et al., 2006).
Alternatively, the range in paleolatitudes presented here demonstrate the large degree of uncertainty with regards to Early
Cretaceous paleogeographic reconstructions of Asia (Supplementary Material; Table S1), which may be driving Aptian-
Albian proxy-model disparities.

**4.4 Atmospheric paleo-$p$CO$_2$**

Cotton and Sheldon (2012) refine procedural guidelines previously established by Cerling and Quade (1993) and Ekart et al.
(1999) for use of pedogenic carbonates in reconstructions of atmospheric $p$CO$_2$ which include maximum limits for $\Delta^{13}C$ (i.e.,
$\delta^{13}C_{carb} - \delta^{13}C_{org}$), isotopic heterogeneity and $\delta^{13}C$ versus $\delta^{18}O$ covariation. They suggest limiting proxy application to
samples with 14‰ < $\Delta^{13}C$ < 17‰ as modern soils with large $\Delta^{13}C$ tend to have S(z) values which fall off of the MAP versus
S(z) relationship defined by Cotton and Sheldon (2012) and are likely to have been disproportionately influenced by
atmospheric $\delta^{13}C$. For our atmospheric $p$CO$_2$ reconstruction, we occluded samples with large $\Delta^{13}C$ (i.e., samples with $\Delta^{13}C$ >
18‰; sample 3B-021). We include two samples in our reconstruction (samples 6-003 and 4-038) which have 17‰ < $\Delta^{13}C$ <
18‰ (Table 1). Though this $\Delta^{13}C$ signature may indicate low-productivity (Cotton and Sheldon, 2012) which can influence
the MAP versus S(z) relationship, the presence of abundant root traces in sections 4 and 6 (i.e., Suarez et al., 2018) suggests
otherwise. In addition to meeting $\Delta^{13}C$ criteria, no clear correlation between carbonate $\delta^{13}C$ and $\delta^{18}O$ is observed (Fig. 4) and
carbonates tend to be isotopically homogeneous (Fig. 4; Table 4; maximum 1$\sigma$ of 0.3‰ in all samples for both $\delta^{13}C$ and
$\delta^{18}O$). We include two estimates of uncertainty in our atmospheric $p$CO$_2$ reconstructions to illustrate the influence of S(z)
estimates on $p$CO$_2$: 1) error bars which represent 1$\sigma$ uncertainty in $\delta^{13}C_{carb}$ and $\Delta_{47}$-based temperatures, and 2) an error
envelop which encompasses the prior uncertainty listed in (1) in addition to a range in S(z) for all samples (Table 4). The
maximum range in S(z) is set using the relationship of Cotton and Sheldon (2012), applying the maximum MAP value
observed in the sections containing samples for $p$CO$_2$ reconstruction (i.e., 984 mm/yr which translates to S(z) of 5309
ppmv). This maximum value is representative of some maximum modern S(z) values observed in Holocene calcic soils by
Breecker et al. (2010) and consistent with summer minimum S(z) values observed in vertisol grasslands by Mielnick and
Dugas (2000). Minimum S(z) is set at 2500 ppmv, following the recommended S(z) of Breecker et al. (2010), as this value is
consistent with minimum MAP for our record following the relationship of Cotton and Sheldon (2012). As observed
previously for the Cretaceous (e.g., Franks et al., 2014), atmospheric paleo-$p$CO$_2$ derived from pedogenic carbonate stable
isotopes tends to lose sensitivity at low atmospheric CO$_2$ concentrations resulting in calculated error which spans negative
concentrations. Here, we exclude negative, unrealistic $p$CO$_2$ values from our record and report these minimums as 0 ppmv
(Fig. 5) and note that calculated minimum $p$CO$_2$ is > −165 ppmv for all samples (Table 4).

Our atmospheric $p$CO$_2$ reconstruction suggests relatively low (for greenhouse climates) and variable $p$CO$_2$ over the study
interval. This observation is consistent with cool Aptian-Albian temperatures (i.e., MAAT ~15 °C in midlatitudes as
indicated by this study and others). Mean atmospheric $p$CO$_2$ for the entire record is 396 ppmv and $p$CO$_2$ generally tracks





temperature variability with low (i.e., < 300 ppmv) $p$CO$_2$ in the cool C10 interval ramping up section to ~1100 ppmv. Our record is largely in agreement with paleobotanical proxy-based $p$CO$_2$ reconstructions for the Aptian-Albian, which range from ~500 to 1300 ppmv (Du et al., 2016; Haworth et al., 2010; Passalia, 2009; Aucour et al., 2008). While this study

indicates slightly lower $p$CO$_2$ than other carbon-isotope based records for the Aptian-Albian (e.g., Ekart et al. (1999) suggest ~1500 ppmv; Wallmann (2001) suggest 700 to 1500 ppmv; Fletcher et al. (2005) suggest 1100 to 1200 ppmv), these records do not all satisfy requirements of Cotton and Sheldon (2012) (e.g., $\Delta^{13}$C < 17‰ in the record of Ekart et al. (1999) likely biases to higher atmospheric $p$CO$_2$), and may lack the sampling resolution to pick up on shorter-term variations. Additionally, though comparatively offset to lower values, variability in our atmospheric $p$CO$_2$ reconstruction follows the

pattern of Aptian-Albian $p$CO$_2$ variability observed in other pedogenic and pelagic marine carbonate-based estimates (i.e., gradual decrease in late Aptian with a low at the Aptian-Albian boundary before increasing into the early Albian; Li et al., 2013; Bottini et al., 2015).

### 4.5 Aptian-Albian variations in atmospheric $p$CO$_2$, climate and the hydrologic cycle

Cooler midlatitude terrestrial temperatures (MAATs of ~15 ºC) are consistent with the post-OAE 1a "cold snap" hypothesis

(e.g., Mutterlose et al., 2009) observed in terrestrial (e.g., Amiot et al., 2011) and sea surface temperature records (e.g., both nannofossil indicators and organic GDGT temperature proxy TEX$_{86}$ show cooling at globally distributed sites; Bottini et al., 2015). Following Friedman and O'Neil (1977), coupled carbonate $\delta^{18}$O and $\Delta_{47}$ suggest variations in $\delta^{18}$O$_{mw}$ of ± 2.2‰ over the study interval consistent with shifts in the distribution of atmospheric moisture associated with climate change. Hence, our $\Delta_{47}$-based temperature estimates prove useful in interpreting regional climate variability over the interval independent of

$\delta^{18}$O$_{mw}$. Climate change-induced variations in Asian continental interior $\delta^{18}$O$_{mw}$ during the early Cretaceous would be expected given model results which show shifts in $\delta^{18}$O$_{mw}$ of +2 to +4‰ associated with two doublings of atmospheric $p$CO$_2$ (global average surface warming of 6 ºC) in continental interiors (Poulsen et al., 2007). Our records similarly indicate warming-induced $^{18}$O-enrichment in $\delta^{18}$O$_{mw}$ for continental Asia, as atmospheric $p$CO$_2$, temperature and $\delta^{18}$O$_{mw}$ increase following the C10 interval (Fig. 5). While age controls are limited, chemostratigraphic correlations (i.e., Suarez et al., 2018)

suggest our record spans several Myr (i.e., C7 to C12 carbon isotope segments after Menegatti et al. (1998) and Bralower et al. (1999); roughly 6 million years). Given the temporal coarseness of our record which likely does not pick up on peak temperature variability, we observe subtle temperature shifts over the interval (i.e., cooling of −2 to −4 °C preceding +4 to +6 °C of warming across the inferred C10 interval), which likely corresponds to the cool interval between OAE 1a and OAE 1b and may include Kilian and/or OAE 1b warmth (e.g., Bottini et al., 2015) around 140 m (Fig. 5).


Shifts in the hydrologic cycle reflected in $\delta^{18}$O$_{mw}$ and MAP (Fig. 5) track $\Delta_{47}$-based temperatures, suggesting climate-controlled regional shifts in interior Asian hydrologic cycle. Interestingly, the driest conditions tend to correlate to relative lows in temperature and $\delta^{18}$O$_{mw}$ perhaps pointing to variations in the seasonality and/or sourcing of meteoric waters in Asia associated with long-term climate evolution. Compared with background environmental conditions, cooler temperatures





(down to 11.1 °C), drier conditions (MAP < 600 mm/yr), and $^{18}$O-depleted meteoric waters are observed in the C10 interval,
        consistent with models of warming-induced precipitation change for the mid-Cretaceous (e.g., Poulsen et al., 2007) and
        observations of more widespread dry conditions in Asia during cool Cretaceous intervals (e.g., Hasegawa et al., 2012).

        Hasegawa et al. (2012) used sedimentological records from Asia to help constrain potential shifts in the descending limb of
the Hadley cell related to Cretaceous climate change and compared these observations to of Hadley cell circulation
        simulations, concluding that as paleo-$p$CO$_2$ concentration increases, so does the width of the Hadley Cell, but that at ~1000
        ppmv paleo-$p$CO$_2$ and greater, the descending limb of the Hadley cell contracts from 30ºN and ºS to 15ºN and ºS
        reorganizing the distribution of atmospheric water vapor. This hypothesis is further supported by paleoenvironmental
        observations (i.e., shifts in lithology and climate-sensitive fossils associated with changes in aridity; Chumakov, 2004;
Chumakov et al., 1995), and general circulation models which indicate a 30ºN position of the descending limb of the Hadley
        cell during Cretaceous greenhouse warmth (Floegel, 2001). The time bins that Hasegawa et al. (2012) investigates are larger
        than the interval investigated here, but the shifts in atmospheric $p$CO$_2$ encompass the range of hypothesized thresholds for
        shifts in Hadley Cell circulation. The relatively cool and dry conditions and $^{18}$O-depleted meteoric waters during the C10
        interval (potentially associated with the "cold snap"; Mutterlose et al., 2009) may reflect on a shift in climate and Hadley cell
circulation, driven by a decrease in atmospheric $p$CO$_2$ (e.g., Hasegawa et al., 2012). Indeed, other sedimentological evidence
        (e.g., glendonites) provide further support for relatively cold conditions at high northern latitudes associated with this
        interval (Vickers et al., 2019). For our records,, atmospheric $p$CO$_2$ drops during the C10 interval from pre- and post-segment
        values within error of 1000 ppmv, to < 350 ppmv in the C10 interval (Fig. 5; Table 4), well below the critical threshold
        proposed by Hasegawa et al. (2012). As atmospheric $p$CO$_2$ increases from the low in the C10 interval to a peak just after,
precipitation and temperature increases similar to the proposed climate shifts that Hasegawa et al. (2012) suggest, in a
        "supergreenhouse" mode. Variability in $\delta^{18}$O$_{mw}$ and MAP is observed throughout our study interval, however, which may
        indicate either multiple fluctuations in Hadley cell circulation across the interval or background regional variability in the
        early Cretaceous hydrologic cycle in continental interior Asia.

        **5 Conclusions**

In summary, new continental Asia midlatitude multi-proxy records of Aptian-Albian carbon cycle, climate and hydrologic
        cycle suggest cool conditions in early Cretaceous midlatitudes (mean of 14.9 ºC; 35ºN to 48ºN paleolatitude depending on
        applied paleogeographic reconstruction) relative to background Cretaceous greenhouse warmth, consistent with our
        estimated atmospheric $p$CO$_2$ (mean value of 396 ppmv) calculated using carbon isotopes in pedogenic carbonates and
        previous regional MAAT observations (Amiot et al., 2011). Variations in the hydrologic cycle (i.e., decreases in MAP and
$\delta^{18}$O$_{mw}$) are associated with transient cooling (−2 to −4 ºC) during the C10 carbon isotope high, consistent with general
        circulation models which suggest differences in temperature, MAP and $\delta^{18}$O$_{mw}$ similar in magnitude to our observations



associated with one to two doubling(s) (or in terms of cooling, halving(s)) of atmospheric $p\mathrm{CO_2}$. These new paleoclimate parameters may be useful to future climate modeling efforts and for understanding potential variability (cooling and warming; shifts in precipitation) in an otherwise greenhouse climate.

**Acknowledgments**

We are grateful to Bruce Barnett for discussion and analytical assistance in the KPESIL at KU, and Katie Snell and Brett Davidheiser-Kroll for discussion, access and assistance in the CUBES-SIL at CUB for clumped isotope analyses. Thank you to Greg Ludvigson for help with cathodoluminescence and lively discussion. This work was supported by the National Science Foundation (grants NSF EAR1941017 to M. Suarez and NSF EAR1024671 to P. Dodson), and grants to H. You from The Strategic Priority Research Program of Chinese Academy of Sciences (grant XDB260000000) and National Natural Science Foundation of China (grants 41688103 and 41872021).

**Data Availability**

All new data reported herein can be found in the tables, Supplementary Material and Open Science Framework (Harper et al., 2020; DOI: 10.17605/OSF.IO/ZUYHN).

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

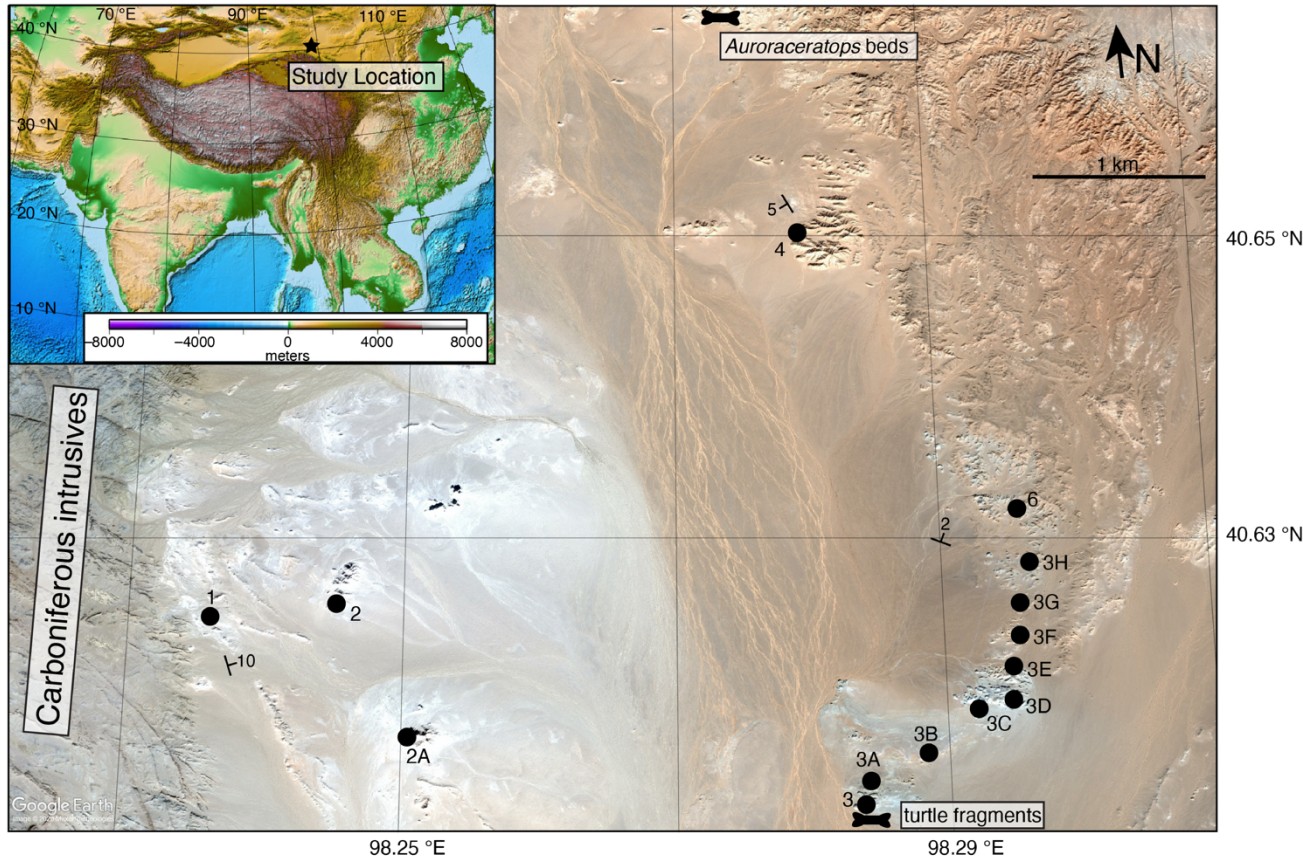


**Figure 1. Field map showing locations of measured sections (circles) and nearby faunal localities (bone symbol) on aerial photography from © Google Earth. Regional modern topographic map of Asia (Amante and Eakins, 2009) inset with study location noted.**



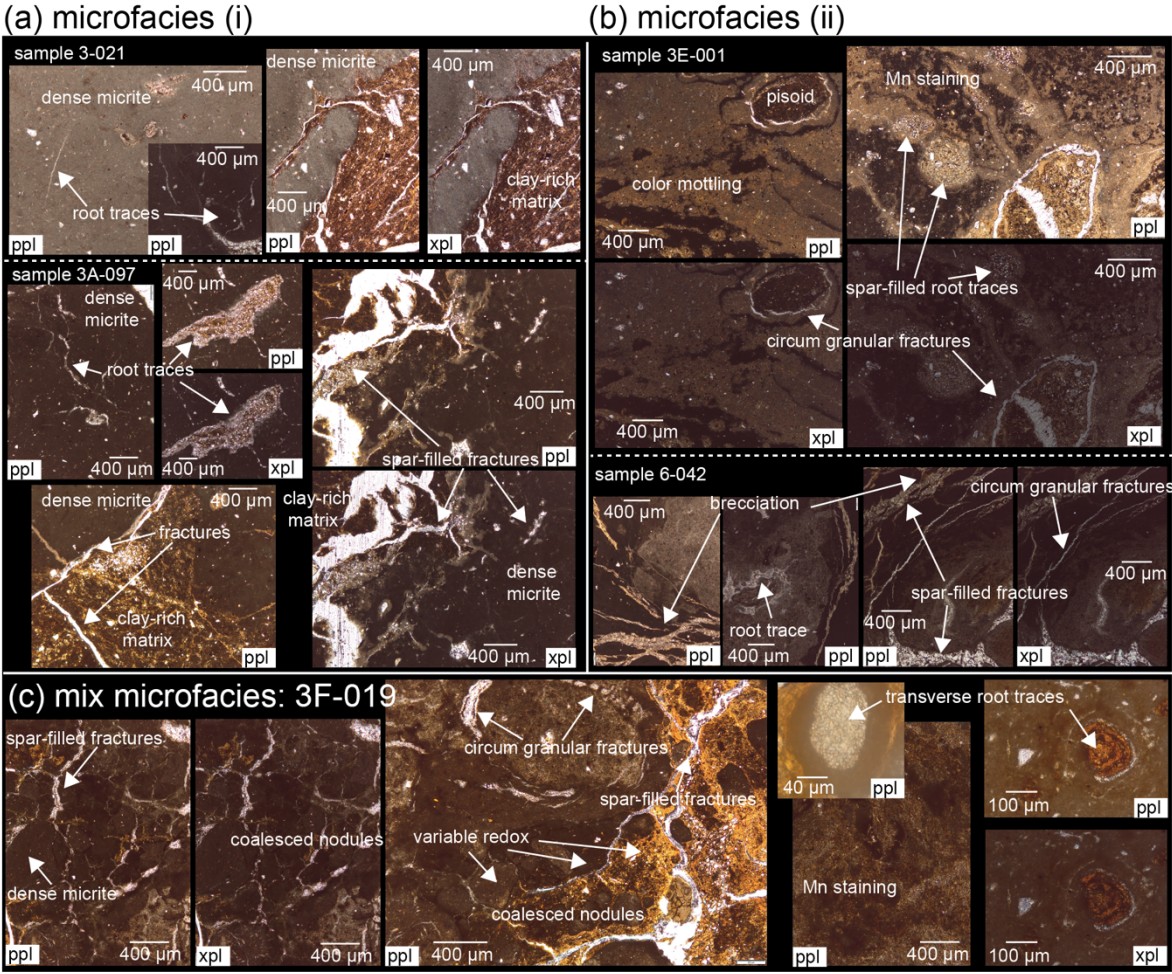

**Figure 2. Annotated photomicrographs from select carbonate nodules of the White Pagoda Site. Samples are split into two microfacies (panels a and b) with sample 3F-019 appearing to be a mixture of the two microfacies (panel c).**




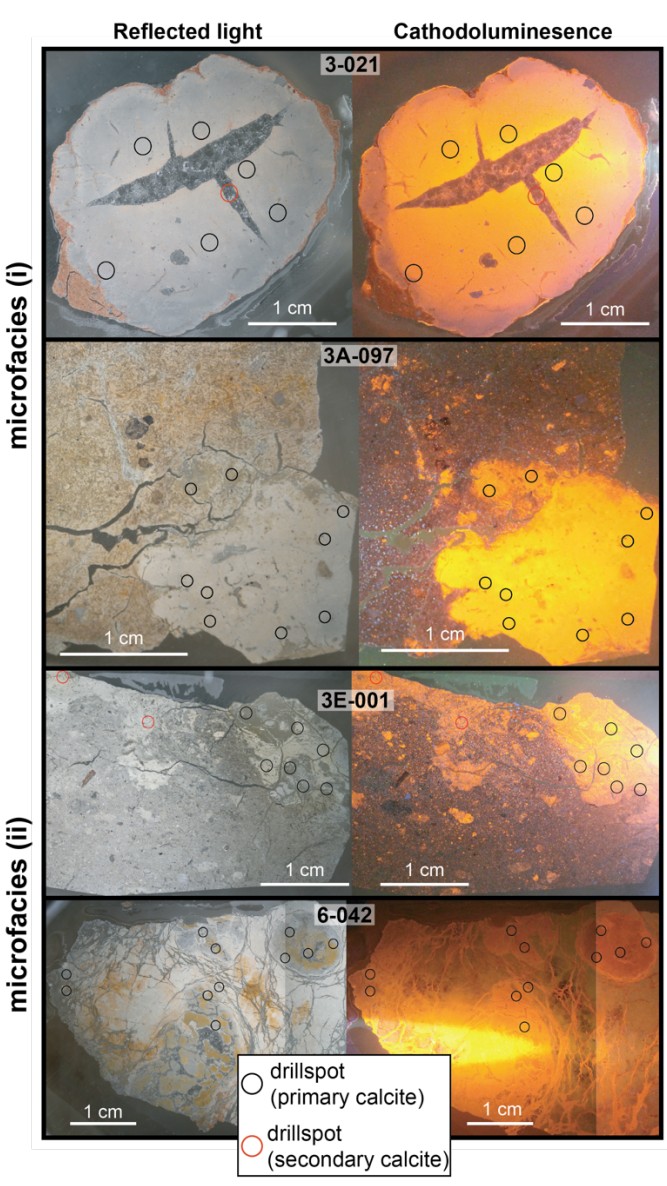

**Figure 3. Reflected light (left) and cathodoluminesence (CL; right) images of select nodule thin sections (two from each microfacies group; top two samples are microfacies (i) and bottom two samples are microfacies (ii)). Drillspots for δ¹³C and δ¹⁸O analysis, mapped from billets, are indicated by circles.**




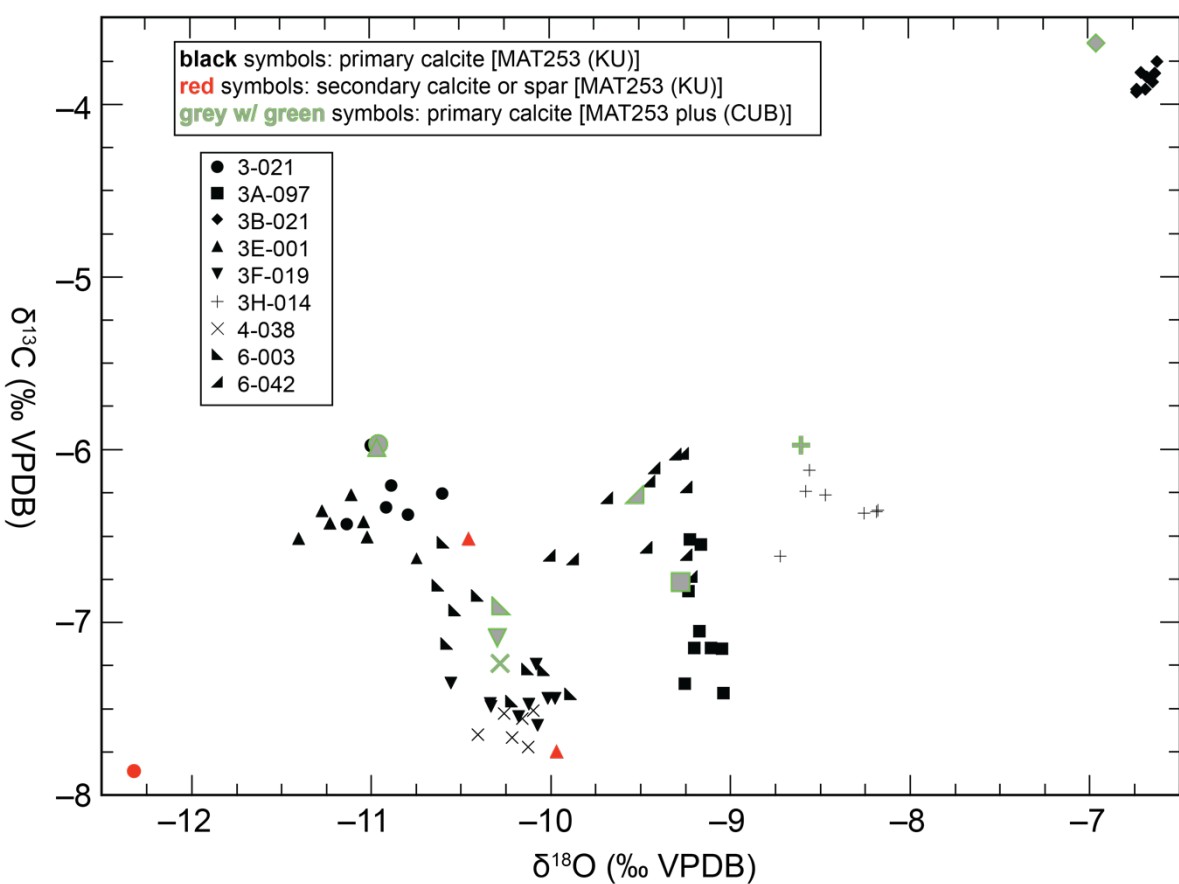

**Figure 4. Carbonate δ¹³C and δ¹⁸O for White Pagoda samples. Black symbols represent drillspot measurements of inferred primary calcite and red symbols represent drillspot measurements of inferred secondary calcite (measured at KU). Grey with green symbols represent mean δ¹³C and δ¹⁸O measured on larger inferred primary calcite samples at CUB.**





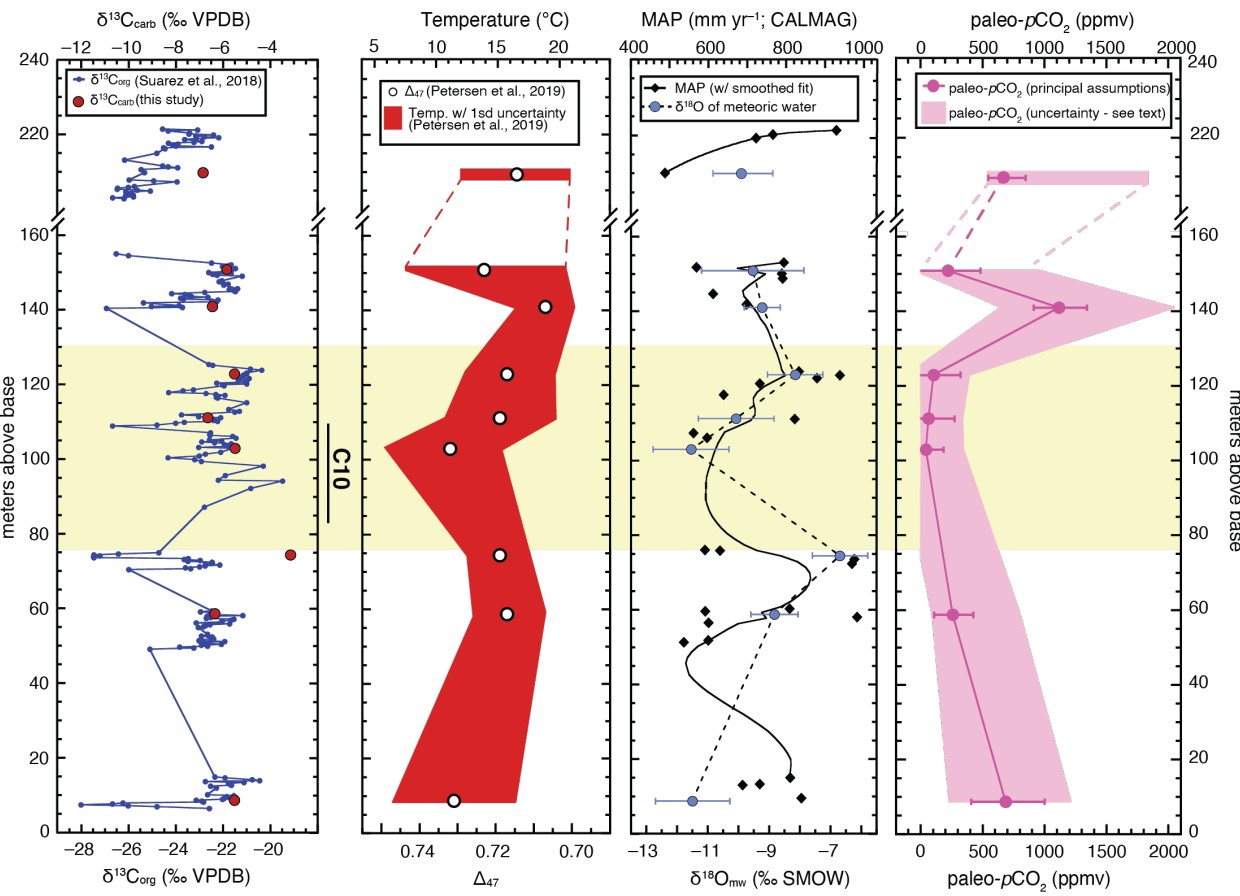

**Figure 5. Multiproxy climate records and record interpretations from the White Pagoda Site, including δ¹³C$_{org}$ of Suarez et al. (2018) and our new records of carbonate δ¹³C, Δ$_{47}$ and temperature (with 1σ uncertainty), MAP and δ¹⁸O$_{mw}$ (with 1σ uncertainty), and atmospheric paleo-$p$CO$_2$ (with 1σ and additional uncertainty considerations; non-positive calculated values are not displayed). The C10 interval has been highlighted. Note the break in depth scale at ~165 meters.**






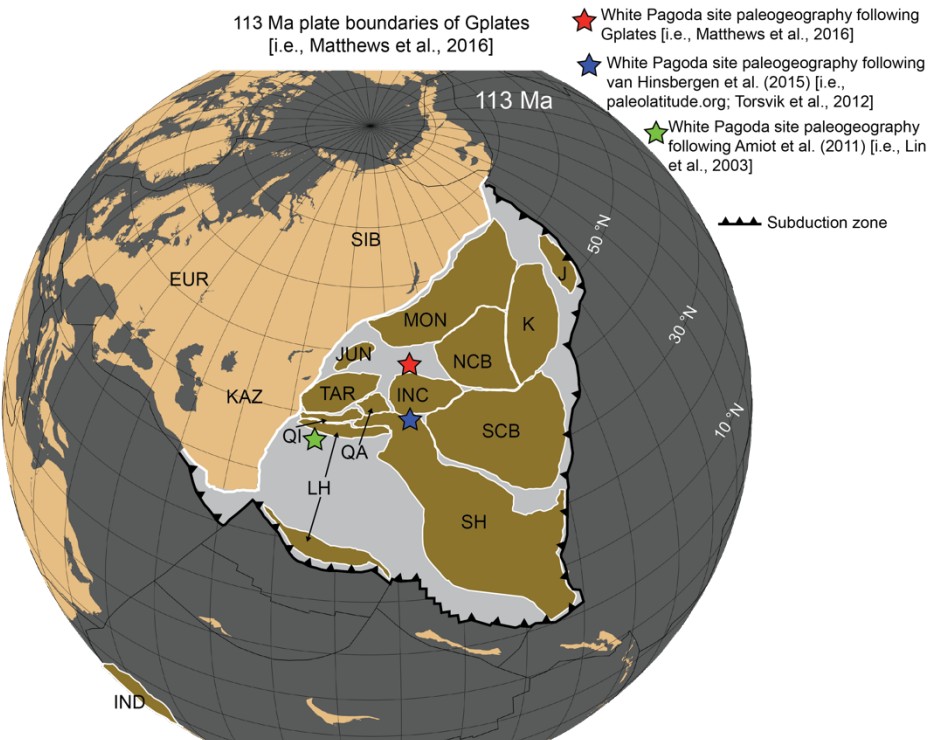

**Figure 6. Paleogeographic reconstruction of plate boundaries ca. 113 Ma (Matthews et al., 2016) using Gplates software. Approximate location of the White Pagoda site following Matthews et al. (2016) is plotted in red. Additional paleogeographic reconstructions are also plotted for comparison (Lin et al., 2003; Torsvik et al., 2012), though these paleolocations are inconsistent with plate boundary reconstructions shown here (i.e., do not follow Matthews et al. (2016)).**






**Figure 7. Northern hemisphere latitudinal gradients of temperature and meteoric water δ$^{18}$O for the mid-Cretaceous (Barremian-Albian). Our mean record data (pink) and the data of Amiot et al. (2011) (black) are plotted with a range in modeled gradients of Cretaceous climate as a function of latitude (blue shaded area) compiled in Suarez et al. (2011a). Details on models included in this range are in main text. Modern temperature and meteoric water δ$^{18}$O (grey lines) are plotted following Rozanski et al. (1993). 1σ uncertainty is included for all data. Two paleolatitude reconstructions are considered: Lin et al. (2003) (panels a and b) and Matthews et al. (2016) (panels c and d). Uncertainty in latitude is included in panels a and b; reconstructions of Matthews et al. (2016) do not provide uncertainty.**



**Table 1. Primary carbonate stable isotope data measured at KU ($\delta^{13}$C and $\delta^{18}$O) and CU Boulder ($\Delta_{47}$, $\delta^{13}$C and $\delta^{18}$O) for White Pagoda Site. $\Delta\delta^{13}$C calculated with $\delta^{13}$C$_{org}$ of Suarez et al. (2018).**

| Sample | Composite section (meters above base) | microfacies | n | $\delta^{13}$C mean (‰ VPDB) | $\delta^{18}$O mean (‰ VPDB) | $\delta^{13}$C 1sd (‰ VPDB) | $\delta^{18}$O 1sd (‰ VPDB) | $\delta^{13}$C$_{org}$ (‰ VPDB) | $\Delta\delta^{13}$C ($\delta^{13}$C$_{carb}$- $\delta^{13}$C$_{org}$) (‰ VPDB) |
|---|---|---|---|---|---|---|---|---|---|
| 3-021 | 8.75 | (i) | 6 | −6.26 | −10.89 | 0.16 | 0.18 | −23.14 | 16.65 |
| 3A-097 | 58.75 | (i) | 9 | −7.02 | −9.16 | 0.32 | 0.08 | −22.55 | 15.53 |
| 3B-021 | 74.50 | (ii) | 8 | −3.86 | −6.69 | 0.06 | 0.05 | −27.48 | 23.62 |
| 3E-001 | 103.00 | (ii) | 7 | −6.45 | −11.12 | 0.12 | 0.21 | −21.59 | 14.99 |
| 3F-019 | 111.25 | mix of (i) & (ii) | 9 | −7.45 | −10.19 | 0.10 | 0.19 | −22.11 | 14.66 |
| 3H-014 | 123.00 | (i) | 7 | −6.33 | −8.43 | 0.15 | 0.22 | −21.00 | 14.67 |
| 6-003 | 141.00 | (ii) | 9 | −7.07 | −10.35 | 0.31 | 0.28 | −25.04 | 17.97 |
| 6-042 | 150.90 | (ii) | 11 | −6.36 | −9.46 | 0.27 | 0.27 | −21.97 | 15.61 |
| 4-038 | 209.85 | (i) | 6 | −7.61 | −10.21 | 0.09 | 0.11 | −25.33 | 17.72 |


**Table 2. Primary carbonate stable isotope data measured at CU Boulder ($\delta^{13}$C and $\delta^{18}$O) and CU Boulder ($\Delta_{47}$, $\delta^{13}$C and $\delta^{18}$O) for White Pagoda Site. Clumped isotopes and interpreted temperatures follow Petersen et al. (2019). Meteoric water $\delta^{18}$O calculated from $\delta^{18}$O$_{carb}$ (CU Boulder measured) and clumped temperatures following Friedman and O'Neil (1977).**

| Sample | Composite section (meters above base) | n | $\delta^{13}$C mean (‰ VPDB) | $\delta^{18}$O mean (‰ VPDB) | $\delta^{13}$C 1sd (‰ VPDB) | $\delta^{18}$O 1sd (‰ VPDB) | $\Delta_{47}$ mean (‰) | $\Delta_{47}$ 1sd (‰) | T (ºC) | T 1sd (ºC) | $\delta^{18}$O$_{water}$ (‰ SMOW) | $\delta^{18}$O$_{water}$ 1sd (‰ SMOW) |
|---|---|---|---|---|---|---|---|---|---|---|---|---|
| 3-021 | 8.75 | 4 | −5.95 | −10.99 | 0.08 | 0.16 | 0.731 | 0.016 | 11.4 | 4.8 | −11.49 | 1.21 |
| 3A-097 | 58.75 | 3 | −6.76 | −9.30 | 0.03 | 0.17 | 0.717 | 0.009 | 15.9 | 2.8 | −8.82 | 0.77 |
| 3B-021 | 74.50 | 4 | −3.67 | −6.98 | 0.04 | 0.40 | 0.719 | 0.007 | 15.0 | 2.3 | −6.69 | 0.90 |
| 3E-001 | 103.00 | 4 | −5.98 | −10.98 | 0.12 | 0.34 | 0.732 | 0.014 | 11.1 | 4.0 | −11.54 | 1.23 |
| 3F-019 | 111.25 | 4 | −7.05 | −10.38 | 0.07 | 0.31 | 0.719 | 0.014 | 15.1 | 4.3 | −10.07 | 1.23 |
| 3H-014 | 123.00 | 4 | −5.95 | −8.63 | 0.08 | 0.17 | 0.717 | 0.011 | 15.9 | 3.5 | −8.15 | 0.91 |
| 6-003 | 141.00 | 4 | −6.86 | −10.32 | 0.02 | 0.13 | 0.707 | 0.007 | 18.8 | 2.2 | −9.22 | 0.59 |
| 6-042 | 150.90 | 4 | −6.27 | −9.60 | 0.02 | 0.30 | 0.723 | 0.021 | 14.0 | 6.3 | −9.53 | 1.66 |
| 4-038 | 209.85 | 4 | −7.24 | −10.32 | 0.04 | 0.13 | 0.715 | 0.014 | 16.5 | 4.4 | −9.71 | 1.07 |








**Table 3. Lithologic and CALMAG (see text for details) data, with interpreted mean annual precipitation (MAP) for White Pagoda Site samples. All samples derived from "B" paleosol horizons.**

| Sample | Composite section (meters above base) | Lithology | CALMAG (%) | MAP (mm/yr; CALMAG) |
|---|---|---|---|---|
| 3-021 | 8.75 | mudstone | 55.4 | 822 |
| 3-035 | 12.25 | mudstone | 47.9 | 651 |
| 3-036 | 12.50 | fine sandstone | 50.1 | 701 |
| 3-043 | 14.25 | mudstone | 54.0 | 790 |
| 3A-069 | 50.50 | mudstone | 40.4 | 481 |
| 3A-071 | 51.00 | mudstone | 43.5 | 552 |
| 3A-085 | 55.75 | mudstone | 43.6 | 553 |
| 3A-091 | 57.25 | mudstone | 62.6 | 984 |
| 3A-097 | 58.75 | mudstone | 43.1 | 543 |
| 3A-100 | 59.50 | mudstone | 54.0 | 788 |
| 3B-009 | 71.50 | mudstone | 61.9 | 969 |
| 3B-014 | 72.75 | mudstone | 62.2 | 976 |
| 3B-023 | 75.00 | mudstone | 45.0 | 586 |
| 3B-024 | 75.25 | mudstone | 43.1 | 542 |
| 3E-010 | 105.25 | mudstone | 43.3 | 548 |
| 3F-001 | 106.50 | mudstone | 41.7 | 509 |
| 3F-015 | 110.25 | mudstone | 54.6 | 803 |
| 3G-003 | 116.75 | mudstone | 45.5 | 596 |
| 3H-001 | 119.75 | mudstone | 50.1 | 701 |
| 3H-007 | 121.25 | mudstone | 57.4 | 867 |
| 3H-010 | 122.00 | mudstone | 60.4 | 934 |
| 3H-014 | 123.00 | claystone | 55.1 | 815 |
| 6-003 | 141.00 | silty mudstone | 48.4 | 663 |
| 6-014 | 143.75 | sandy siltstone | 44.1 | 566 |
| 6-030 | 147.90 | mudstone | 53.0 | 768 |
| 6-035 | 149.15 | mudstone | 52.9 | 765 |
| 6-042 | 150.90 | carb. mudstone | 42.0 | 517 |
| 6-047 | 152.15 | carb. mudstone | 53.2 | 771 |
| 4-042 | 210.85 | mudstone | 40.2 | 476 |
| 4-077 | 219.95 | mudstone | 49.9 | 697 |
| 4-081 | 220.95 | muddy sandstone | 51.7 | 738 |
| 4-085 | 221.95 | muddy sandstone | 58.5 | 892 |


**Table 4. Atmospheric $pCO_2$ calculated using $\delta^{13}C_s$ (from $\delta^{13}C_{carb}$ with temperature-dependent enrichment factor; Romanek et al., 1992); 1 sd uncertainty in $\delta^{13}C_{carb}$ and temperature propagated to compute uncertainty in $pCO_2$ ($pCO_2$ + and $pCO_2$ −); $pCO_2$ +\* and $pCO_2$ −\* indicate prior propagated uncertainty and uncertainty in S(z) [see text]. $\delta^{13}C_a$ = average Atlantic bulk carbonate for C isotope segments following Bralower et al. (1999) with DIC-atmosphere fractionation of −8.23‰). $\delta^{13}C_r$ = organic C record of Suarez et al. (2018). S(z) is estimated from MAP following Cotton and Sheldon (2012). All $\delta^{13}C$ values reported are relative to VPDB.**


| Sample | Composite section (meters above base) | $\delta^{13}C_s$ (‰) | $\delta^{13}C_s$ unc. (+/- ‰) | C isotope segment | $\delta^{13}C_a$ (‰) | $\delta^{13}C_r$ (‰) | S(z) (MAP-derived) | $pCO_2$ (ppmv) | $pCO_2$ + (ppmv) | $pCO_2$ − (ppmv) | $pCO_2$ +\* (ppmv) | $pCO_2$ −\* (ppmv) |
|---|---|---|---|---|---|---|---|---|---|---|---|---|
| 3-021 | 8.75 | −16.87 | 0.73 | C6 | −4.3 | −23.14 | 4393 | 686 | 314 | 280 | 523 | 455 |
| 3A-097 | 58.75 | −17.09 | 0.65 | C8 | −4.4 | −22.55 | 2810 | 257 | 167 | 150 | 543 | 162 |
| 3E-001 | 103.00 | −17.09 | 0.60 | C10 | −4.2 | −21.59 | 2835 | 42 | 141 | 129 | 301 | 204 |
| 3F-019 | 111.25 | −17.62 | 0.62 | C10 | −4.3 | −22.11 | 4281 | 62 | 211 | 192 | 277 | 224 |
| 3H-014 | 123.00 | −16.40 | 0.57 | C11 | −4.2 | −21.00 | 4353 | 103 | 217 | 198 | 288 | 219 |
| 6-003 | 141.00 | −16.79 | 0.58 | C12 | −4.4 | −25.04 | 3488 | 1116 | 225 | 205 | 925 | 463 |
| 6-042 | 150.90 | −16.66 | 1.02 | C12 | −4.4 | −21.97 | 2661 | 218 | 261 | 221 | 737 | 224 |
| 4-038 | 209.85 | −17.61 | 0.61 | C15 | −5.4 | −25.33 | 2427 | 682 | 164 | 148 | 1168 | 132 |