# Peer review of "Aptian-Albian clumped isotopes from northwest China: Cool temperatures, variable atmospheric $pCO_2$ and regional shifts in hydrologic cycle"

_Climate of the Past, 2020_

## Referee Comment (RC1) · Anonymous Referee #1 · 25 Jan 2021

The manuscript entitled as "Aptian-Albian clumped isotopes from northwest China: Cool temperatures, variable atmospheric pCO2 and regional shifts in hydrologic cycle" by Dustin T. Harper et al., present new results of pedogenic carbonate stable isotopes ($\delta$13C, $\delta$18O and $\Delta$47) from the lower mid-Cretaceous (Aptian-Albian) Xiagou and Zhonggou formations, Yujingzi Basin of NW China. Authors estimate the MAT using $\Delta$47 of carbon ad oxygen isotopic values and the MAP using CALMAG of mudrocks, further calculate the pCO2 using MAP and $\delta$13C of pedogenic carbonates with other parameters, and discuss the carbon and hydrologic cycles for the interval of the

[Figure]

Aptian-Albian. This work is a new progress of the land quantitive paleoclimate in North China, even in East Asia. It would provide clues and references for the climatic reconstruction of the greenhouse Cretaceous period. However, more data and evidence need to further enhance and refine, and some issues have to solve.

1) Geological data to complement. Though figure 1 shows the locations of samples in outcrop, it does not have any geological significances. It does not exhibit any stratigraphic sequences with sample horizons. In my opinion, it is important that sampling locations are plotted in a geological sketch. And it is advised that a sampling log is added.

2) Data age. Authors used the organic stable carbon isotope chemostratigraphic records for the site (Suarez et al., 2018) to have the studied strata age-controlled. Albeit more than 400 organic stable carbon isotopes were correlated and suggested the Aptian-Albian in Suarez et al. (2018), it is not assured and persuaded due to lack of precise age reference-point and age-index fossils. This is a common issue and problem of the land materials for paleoclimatic analysis. It is cautious to make the precise correlation for the terrestrial strata and samples.

3) Field description of paleosols. It is key to make sure the observed horizons are real paleosols, i.e. authors claimed the paleosols a kind of vertisol. I do not see the details of the paleosols even though authors have done analyses of CL and microfacies for the calcareous nodules. Shape, size, content, and occurrence of nodules can provide us reference for paleosols. Color, structures, and ped types can give us some information about the paleosols. Other more, vertisols are a kind of paleosols we do not easily observe and distinguish in practice in field recognition. Detailed notation of evidence seems necessary for the classification of the vertisol.

4) Drilling samples for clumped isotopes. It is a good job for the clumped isotope. But it is also a problem to take powder samples from the calcretes. This is because we only need <0.1 mg for the common C-O analyses of carbonates, but we have to take over 5

mg for clumped isotope analysis. It is difficult to take so much from a calcretes sample according to much experience. So, how to get the enough quantity of the powder sample may need to expain.

5) Low temperature and low pCO2 in the Aptian-Albian. As we know from lots paleoceanographic and paleoclimatic achievements, the mid-Cretaceous is the hottest interval in the Phanerozoic. So, the conclusion from the authors that a low temperature had been in the Aptian-Albian may need to further examine except for the short "cold snap". It may be paradox that low temperature is consistent with low pCO2 in the Aptian-Albian in Northwest China. This is because the pCO2 is almost global in nature, but the temperature in a basin, North China, was probably local record to great degree on land. Actually, we know the Cretaceous climate was not homogeneous in China.

6) Some references are not regularly lined in text. For examples, references cited in Lines 55, 269, etc. are neither listed in the sequence of publishing time nor of surname letter.

---

## Referee Comment (RC2) · Anonymous Referee #2 · 24 Feb 2021

Harper et al. use pedogenic carbonates and paleosol elemental geochemistry to develop new temperature, precipitation, and pCO2 estimates of "mid" Cretaceous paleoclimate in northwest China. They use these records to confirm previous reconstructions, and to suggest that these conditions may represent examples of thresholds in shifting Hadley circulation at this time.

Overall this work contributes useful new data for the region and time period, but could benefit from a refocus of the work within geographic context and with additional discussion of regional climate and potential uncertainties. Below are some comments on

particular aspects of the work that could be improved or reevaluated before publication.

Comments and revisions:

1) Improve editing of the manuscript throughout (incorrect agreement, missing words, etc.).

2) Is the -8.23 per mil correction for  $\delta$ 13Ca reasonable for this period, given the existence of glendonites and the low temperatures and low pCO2? Cooler, low-pCO2 periods during the Cenozoic have substantially higher  $\delta$ 13Ca values (-5.5 to -6.5 per mil; Tipple et al., 2010), which may change your eventual pCO2 estimates (make them slightly higher?).

3) CALMAG is an elemental ratio, and should not be reported in % (e.g., Nordt et al., 2010).

4) These cathodoluminescence images are concerning. High luminescence indicates substantial Mn, Fe, etc. which is usually indicative of diagenesis (e.g., Driese & Mora, 1993; Budd et al., 2002), which appears to be what you sampled. Also, the final image (Figure 3, sample 6-042) is incorrectly illuminated and the bright region is just showing an incident beam from the CL (which is not calibrated across the surface). You may want to reevaluate your data to distinguish between samples selected from different regions of the carbonate nodules, and confirm that the presented data are from primary materials.

5) Why do you need Figure 6 showing different paleogeographies? Unless you add in simulations of MAT and  $\delta$ 18Omw as an overlay (e.g., Zhou et al., 2008; Hasagawa et al., 2012), this doesn't really contribute to the paper. Instead just rely on Figure 7 to show what you're arguing with respect to paleogeography, and perhaps expand the discussion of this point to match.

6) CALMAG values reported are different in the Results vs. the Discussion- does the version in the Discussion and in Figure 5 include non-B horizons? Check this and

CPD
revise (or specify) as needed. The CALMAG-derived MAP fit in Figure 5 is also overly smoothed- there are not enough data points for the level of smoothing (moving average I assume?), which results in data artefacts like the curve at  $\sim$ 40m.

7) You are reporting false precision in  $\delta$ 180mw and temperature (and raw data tables too)- edit this to reflect precision within reported uncertainties.

8) Your highest pCO2 values come from samples outside the "accepted"  $\Delta$ 13C range for this proxy (e.g., Cotton and Sheldon, 2012). As a result, perhaps all of your estimated values suggest low pCO2 for this period (

uncertainties in paleotemperature estimates (e.g., Fernandez et al., 2017; Bernasconi et al., 2021).

---

## Referee Comment (RC3) · Hilary Corlett (Referee) · 6 Mar 2021

I was asked to provide a review focused on the reviewer comment below and Figure 3 from cp-2020-152.

The reviewer comment is: These cathodoluminescence images are concerning. High luminescence indicates substantial Mn, Fe, etc. which is usually indicative of diagenesis (e.g., Driese & Mora, 1993; Budd et al., 2002), which appears to be what you sampled. Also, the final image (Figure 3, sample 6-042) is incorrectly illuminated and

the bright region is just showing an incident beam from the CL (which is not calibrated across the surface). You may want to reevaluate your data to distinguish between samples selected from different regions of the carbonate nodules, and confirm that the presented data are from primary materials.

Please find my review below: I would agree that the images are concerning. They looked odd to me and when I read the text, the method of "Macroscale imaging through the 50 mm top window of the chamber was carried out using a 16 Mpx Canon EOS SL1 DSLR camera with a macro lens suspended over the CL chamber" may explain why the luminescence is not what you expect from a standard CL image. These images would normally be taken with a C-mounted microscope camera or C-mount DSLR, or if C-mount is missing, then you would use an ocular mounted USB microscope camera or DSLR. The luminescence in the photos may be exaggerated in some way, which may be why the reviewer is concerned. Also, the review is correct about the last image where it is showing the incident beam. This image should not be used.

The cause of luminescence is Mn, and Fe is more of a quenching element. If these nodules formed under slightly reduced conditions, you would expect there to be some uniform, dull orange luminescence. In a pedogenic environment, you may expect there to be some luminescence because pore waters forming these nodules may be relatively Fe poor, if the Fe is oxidized, as is the case in these types of environments. I wouldn't agree (with the reviewer's comment) that any luminescence at all means diagenesis. Given the fact that the zones indicated as "primary" have a uniform luminescence in the first two pictures, as long as the isotope data is fairly consistent (i.e. showing several data points in a narrow range), I wouldn't say these have been diagenetically overprinted. The third image has a more mottled appearance and therefore, may have experienced some diagenesis. It is impossible to say anything about the fourth image because it is just showing the incident beam and causing one area to be more brightly illuminated than the surrounding. The best example here is the first image with a uniform orange/yellow luminescence of the nodule, marked as primary, and the fracture

with mottled appearance marked as secondary.

The final word from me would be: 1) get rid of the last image, 2) mention that the method of photography used here may have resulted in overexposure of the CL images, and 3) only images 1 and 2 are convincing as "primary" formed from slightly reduced water, enriched in Mn. If the third nodule also resulted in isotope data that falls within the range of the first two, then this mottled appearance may just indicate this nodule has experienced minor diagenetic alteration or when it formed, it incorporated some of the matrix into the nodule, giving it a less uniform luminescence.

For context, I am a carbonate sedimentologist with a primary research focus on diagenesis. I am familiar with CL of carbonates but have not read the rest of the paper as I was only asked to provide this limited review. The review process required me to fill in the recommendations, but please note that my recommendation is limited to the CL imagery and interpretation. For all of my other recommendations, I defaulted to the median.

Hilary Corlett

---

## Author Comment (AC1) · 6 Apr 2021

General Response to Reviewer Commentary

First, the authors of the manuscript entitled "Aptian-Albian clumped isotopes from northwest China: Cool temperatures, variable atmospheric pCO2 and regional shifts in hydrologic cycle" would like to thank the three reviewers for providing focused critical evaluations of our work. Below, we directly address the reviewer's comments and include the original reviewer commentary. Original reviewer comments are labeled

"RC1", "RC2" and "RC3". We intend on largely following the advice and comments of reviewers. In our responses, we layout specific planned revisions for the next draft submission. We are confident that we can address the reviewer's concerns with only these few, minor revisions to the manuscript. We look forward to hearing the editor's decision for the next stage of the manuscript.

Response to Review 1 (Anonymous)

RC1: The manuscript entitled as "Aptian-Albian clumped isotopes from northwest China: Cool temperatures, variable atmospheric pCO2 and regional shifts in hydro-logic cy- cle" by Dustin T. Harper et al., present new results of pedogenic carbonate stable isotopes ($\delta$13C, $\delta$18O and $\Delta$47) from the lower mid-Cretaceous (Aptian-Albian) Xiagou and Zhonggou formations, Yujingzi Basin of NW China. Authors estimate the MAT using $\Delta$47 of carbon ad oxygen isotopic values and the MAP using CALMAG of mu- drocks, further calculate the pCO2 using MAP and $\delta$13C of pedogenic carbonates with other parameters, and discuss the carbon and hydrologic cycles for the interval of the Aptian-Albian. This work is a new progress of the land quantitive paleoclimate in North China, even in East Asia. It would provide clues and references for the cli-matic re- construction of the greenhouse Cretaceous period. However, more data and evidence need to further enhance and refine, and some issues have to solve.

Geological data to complement. Though figure 1 shows the locations of samples in outcrop, it does not have any geological significances. It does not exhibit any strati-graphic sequences with sample horizons. In my opinion, it is important that sampling locations are plotted in a geological sketch. And it is advised that a sampling log is added.

Author Response: Much of the geologic data and sampling information for the study site was initially described in Suarez et al. (2018). We have also included supplemental tables in the present study detailing sample lithology and samples identified as pale-osol B-horizons. However, to address the reviewer's concern for lack of stratigraphic

and geologic data, we opt to include additional supplemental figures in the next draft submission which will include lithostratigraphic columns with sampling locations, and images of sampled carbonate nodules, nodule features and sedimentary structures interpreted to represent pedogenesis (slickensides, ped structure etc.).

RC1: Data age. Authors used the organic stable carbon isotope chemostratigraphic records for the site (Suarez et al., 2018) to have the studied strata age-controlled. Albeit more than 400 organic stable carbon isotopes were correlated and suggested the Aptian-Albian in Suarez et al. (2018), it is not assured and persuaded due to lack of precise age reference-point and age-index fossils. This is a common issue and problem of the land materials for paleoclimatic analysis. It is cautious to make the precise correlation for the terrestrial strata and samples.

Author Response: While we agree chemostratigraphic correlation can be complicated by local influences over global variations in carbon cycle, we argue that Suarez et al. (2018) provide convincing evidence that our study sections span the "C10" carbon isotope high after Menegatti et al. (1998) and Bralower et al. (1999). Variations in $\delta$13C of organic C from the same sample collection utilized in this study have been correlated to globally representative records of carbon cycle variations.

Further, this is not the only age constraint for the Xinminpu Group. Samples from localities to the South of the Yujingzi basin have been identified as Xinminpu Group and measured for radiometric dates. These dates range from 123 $\pm$2.6 Ma to 113.7 $\pm$1.8 Ma (Kuang et al., 2013; Li et al., 2013).

Additionally, a recent study (Zheng et al., 2021) establishes regional ages through bio- and chronostratigraphy. This study reviews available age controls for the Lower Cretaceous in NW China and places the organic carbon isotope records of Suarez et al. (2018) (i.e., our study sections) within the chronostratigraphic framework of the region. Now that this paper has been published, we will include more details of the biostratigraphy and chronostratigraphy which better establish the age control of our

study site in the next draft submission.

RC1: Field description of paleosols. It is key to make sure the observed horizons are real paleosols, i.e. authors claimed the paleosols a kind of vertisol. I do not see the details of the paleosols even though authors have done analyses of CL and microfacies for the calcareous nodules. Shape, size, content, and occurrence of nodules can provide us reference for paleosols. Color, structures, and ped types can give us some information about the paleosols. Other more, vertisols are a kind of paleosols we do not easily observe and distinguish in practice in field recognition. Detailed notation of evidence seems necessary for the classification of the vertisol.

Author Response: Much of this information was established in Suarez et al. (2018) for the study sections, and for this study we avoided redundancies with respect to detailed outcrop and paleosol descriptions. However, we understand the importance of establishing the sampled paleosols as vertisols to interpretation of our results. Following this reviewer's comment, the authors have decided to include an additional supplemental figure (also described above) which will include strat columns with descriptive observations of pedogenic structures and images of hand samples. Additionally, we plan to update our field map based on this review and review 2. This site map will show the region with localities of geologic and paleontologic interest as well as select outcrop images which show, for example, mukkara cracks and other evidence supporting our classification.

RC1: Drilling samples for clumped isotopes. It is a good job for the clumped isotope. But it is also a problem to take powder samples from the calcretes. This is because we only need <0.1 mg for the common C-O analyses of carbonates, but we have to take over 5mg for clumped isotope analysis. It is difficult to take so much from a calcretes sample according to much experience. So, how to get the enough quantity of the powder sample may need to expain.

Author Response: Our pedogenic carbonate nodules were cm to multi-cm in scale,

providing ample carbonate material for sampling. The reviewer makes a good point in that it is important that the clumped isotope material that is sampled should maintain uniform $\delta$13C and $\delta$18O values. To ensure this, we measured multiple "spot" samples for $\delta$13C and $\delta$18O for each nodule, and only sampled material for clumped isotopes from areas with uniform $\delta$13C and $\delta$18O in a given sample. The average $\delta$13C and $\delta$18O values were then compared with the values obtained during the clumped isotope measurement as an indication of successful sampling. We argue that this strategy has been clearly outlined in the manuscript. In addition, the new supplemental figure will show the sizes of typical carbonate nodules sampled so that readers can have a better sense for the material sampled.

RC1: Low temperature and low pCO2 in the Aptian-Albian. As we know from lots paleoceanographic and paleoclimatic achievements, the mid-Cretaceous is the hottest interval in the Phanerozoic. So, the conclusion from the authors that a low temperature had been in the Aptian-Albian may need to further examine except for the short "cold snap". It may be paradox that low temperature is consistent with low pCO2 in the Aptian-Albian in Northwest China. This is because the pCO2 is almost global in nature, but the temperature in a basin, North China, was probably local record to great degree on land. Actually, we know the Cretaceous climate was not homogeneous in China.

Author Response: The coauthors humbly disagree that the scientific community has concluded that the mid-Cretaceous was the hottest interval of the Phanerozoic. Taken as a whole, yes, the Cretaceous Period was one of the warmest intervals, but this paper and many others investigate a more detailed climate record. At the stage level, many would argue that the hottest interval occurred during the Late Cretaceous near the Cenomanian-Turonian boundary (Bice et al., 2006; Hay et al., 2017). Certainly, we know that the Cretaceous was generally a warm interval (Hay et al., 2017), but a large number of publications which examine shorter-term climate variations indicate relative cool conditions at the Aptian-Albian, but also other stages such as the Valanginian (see for example, Mutterlose et al., 2009; Rodriguez-Lopez 2016; Bottini et al., 2015;

Heimhofer et al., 2008; Millan et al., 2014; Vickers et al., 2019). Undoubtedly, there were large variations in the carbon cycle during this time evidenced by ocean anoxic events and variable $\delta$13C in global archives, suggesting these variations were global in nature (Menegatti et al., 1998; Bralower et al., 1999). Indeed, previous work on mid-Cretaceous climate suggests relatively cool temperatures over a large swath of latitude in Asia (Amiot et al., 2011).

We agree that surface temperatures on Earth are not homogeneous, including over a large land mass such as Asia. Also, we agree that our temperature record reflects a local signal. Atmospheric pCO2 is certainly a global signal when we compare atmospheric CO2 mixing times with the temporal resolution of our record. However, shifts in pCO2 are clearly linked with shifts in land surface temperatures over many intervals (e.g., Bice et al., 2006; Pagani et al., 2005; Hay et al., 2017). In the manuscript we have discussed the local versus global nature of our records, both in terms of topography and with respect to latitude, fairly extensively in the discussion section 4.3 "Latitudinal gradients of temperature and meteoric water $\delta$18O for the Aptian-Albian". In this section, our temperature record was placed within a global context using latitudinal temperature profile figures (Figure 7).

Indeed, one large take-away from this work is that Cretaceous climate was not homogenous neither spatially (see discussion and figures on latitudinal temperature distribution) nor temporally. Variations were clearly occurring in terms of climate and the carbon cycle, evidenced by: 1. Shifts in global records of carbonate $\delta$13C (Menegatti et al. 1998; Bralower et al. 1999; Suarez et al., 2018) 2. Multi-million year records of variable atmospheric pCO2 including ours and those of Bice et al. (2006) and Wang et al. (2014). 3. Mid-Cretaceous records of temperature (terrestrial and marine; for example, Bice et al., 2006; Amiot et al., 2011; O'Brien et al., 2017; this study)

RC1: Some references are not regularly lined in text. For examples, references cited in Lines 55, 269, etc. are neither listed in the sequence of publishing time nor of surname letter.

Author Response: Thank you for pointing this out. We plan to address this in the next draft submission.

---

## Author Comment (AC2) · 6 Apr 2021

General Response to Reviewer Commentary

First, the authors of the manuscript entitled "Aptian-Albian clumped isotopes from northwest China: Cool temperatures, variable atmospheric pCO2 and regional shifts in hydrologic cycle" would like to thank the three reviewers for providing focused critical evaluations of our work. Below, we directly address the reviewer's comments and include the original reviewer commentary. Original reviewer comments are labeled

"RC1," "RC2," and "RC3" below. We intend on largely following the advice and comments of reviewers. In our responses, we layout specific planned revisions for the next draft submission. We are confident that we can address the reviewer's concerns with only these few, minor revisions to the manuscript. We look forward to hearing the editor's decision for the next stage of the manuscript.

Response to Review 2 (Anonymous) RC2: Harper et al. use pedogenic carbonates and paleosol elemental geochemistry to de- velop new temperature, precipitation, and pCO2 estimates of "mid" Cretaceous pale- oclimate in northwest China. They use these records to confirm previous reconstruc- tions, and to suggest that these conditions may represent examples of thresholds in shifting Hadley circulation at this time. Overall this work contributes useful new data for the region and time period, but could benefit from a refocus of the work within geographic context and with additional discussion of regional climate and potential uncertainties. Below are some comments on particular aspects of the work that could be improved or reevaluated before publication. Comments and revisions: 1) Improve editing of the manuscript throughout (incorrect agreement, missing words, etc.).

Author Response: Thank you for pointing this out. We plan to address this in the next draft submission.

RC2: 2) Is the -8.23 per mil correction for $\delta$13Ca reasonable for this period, given the ex- istence of glendonites and the low temperatures and low pCO2? Cooler, low-pCO2 periods during the Cenozoic have substantially higher $\delta$13Ca values (-5.5 to -6.5 per mil; Tipple et al., 2010), which may change your eventual pCO2 estimates (make them slightly higher?).

Author Response: For our study, a $-8.23$ ‰ correction yields $\delta$13Ca values which range from $-5.38$ ‰ to $-4.18$ ‰ (Table 4). These values are indeed higher than those the reviewer lists from Tipple et al., 2010). Perhaps the reviewer was thinking that $-8.23$ ‰ was the applied $\delta$13Ca value for all sample calculations? We can assure you

this is not the case. We will more clearly lay out the applied offset and resulting $\delta$13Ca values in the next draft submission.

If the reviewer was intending on recommending using lower values in the range of the cool Cenozoic values established by Tipple et al. (2010) (i.e., $-6.5$‰ to $-5.5$ ‰, we argue that given the relative differences in climate between the cool intervals of the Cenozoic and the cool intervals of the Cretaceous (i.e., the cool Cretaceous was likely warmer than cool Cenozoic; Hay et al. 2017; Bice et al., 2006; Westerhold et al., 2020), our slightly higher ( $+1.0$‰ $\delta$13Ca values are appropriate. If, however, lower values ( $-6.0$ ‰ were applied, the reviewer is correct in stating that pCO2 estimates would tend to increase marginally.

As an example, if the $\delta$13Ca value for sample 4-038 was adjusted to $-6.0$ ‰ (mid-point in the range suggested by the reviewer), the reconstructed atmospheric pCO2 value would shift from 682 to 712 ppmv.

RC2: 3) CALMAG is an elemental ratio, and should not be reported in

Author Response: Thank you for pointing this out. We plan to address this in the next draft submission.

RC2: 4) These cathodoluminescence images are concerning. High luminescence indicates substantial Mn, Fe, etc. which is usually indicative of diagenesis (e.g., Driese Mora, 1993; Budd et al., 2002), which appears to be what you sampled. Also, the final image (Figure 3, sample 6-042) is incorrectly illuminated and the bright region is just showing an incident beam from the CL (which is not calibrated across the surface). You may want to reevaluate your data to distinguish between samples selected from different regions of the carbonate nodules, and confirm that the presented data are from primary materials.

Author Response: As the third reviewer suggests, these CL images were taken with conditions for high luminescence sensitivity (e.g., He chamber). Pedogenic carbonate

nodules form during post-deposition recrystallization and hence will have features associated with this degree of alteration. High luminescence does not always indicate degree of diagenesis evidenced by the low-luminescence fracture-filled spar. This spar tended to have an isotopic diagenetic signature (lower $\delta$18O and $\delta$13C values) when compared to homogenous isotope values of the highly-luminescent nodule carbonate (Figure 4). In addition, the luminescence may actually be expected in these soils if they were seasonally saturated rather than a specific indication of degree of diagenesis. The Budd et al. (2002) reference that Reviewer 2 recommends describes variable luminescence in addition to discernible disequilibrium between $\delta$13Ccarb and $\delta$13Corg as evidence for diagenetic alteration from environmental values. Our data is rather homogenous in its luminescence and only one sample appears to have carbon isotope values suggestive of disequilibrium. This particular sample was removed from use for calculation of pCO2. However, we are comfortable including the clumped isotope temperature value because clumped isotope derived temperatures are independent of stable isotope values, so even if there was some amount of early diagenesis, it likely still represents near surface temperatures (i.e., pre-burial) as pedogenic carbonates tend to form over thousands of years (Giles et al., 1966).

We acknowledge that sample 6-042 is poorly illuminated and shows the incident beam. To address the sub-optimal quality of CL imaging in the manuscript, we intend on including new images in the next draft submission. These new images will be captured with the aim of addressing the specific reviewer concerns discussed in cathodoluminescence comments of Reviewers 2 and 3.

We do indeed evaluate our data by distinguishing $\delta$13C and $\delta$18O from different regions of the nodules (luminescent nodule vs. spar) in Figure 4 as the reviewer suggests. Generally, apparent secondary calcite phases (e.g., spar) are offset from the ranges in multi-spot stable isotope values of micritic, likely primary, phases of calcite (Figure 4).

RC2: 5) Why do you need Figure 6 showing different paleogeographies? Unless you add in simulations of MAT and $\delta$18Omw as an overlay (e.g., Zhou et al., 2008;

Hasagawa et al., 2012), this doesn't really contribute to the paper. Instead just rely on Figure 7 to show what you're arguing with respect to paleogeography, and perhaps expand the discussion of this point to match.

Author Response: We argue this is an important figure for the manuscript and intend to keep it in the next draft submission for 2 reasons: 1) Allows readers to place the study location within a greater tectonic framework of the time. Including the paleogeography of the study can help readers more clearly understand paleoenvironmental setting and potential complication with regards to our interpretation of paleoenvironment 2) This figure provides a visual aid of ranges of paleolatitude for key sites in Asia for the Aptian-Albian which has not been previously done. Much of the literature which describes Aptian-Albian climate in Asia relies on these reconstructions. These studies may tend to exaggerate cool or warm conditions for a region accordingly if they do not consider all possible reconstructions.

RC2: 6) CALMAG values reported are different in the Results vs. the Discussion- does the version in the Discussion and in Figure 5 include non-B horizons? Check this and revise (or specify) as needed. The CALMAG-derived MAP fit in Figure 5 is also overly smoothed- there are not enough data points for the level of smoothing (moving average I assume?), which results in data artefacts like the curve at âĹij40m.

Author Response: In the results and Supplemental Table S4 we describe and list CAL-MAG values for all available samples. In Table 3 and Figure 5 (as well as in the discussion), we only include paleosol B-horizon samples which are within the range appropriate to apply MAP-calibration following recommendations by Nordt and Driess (2010). Please see section 3.3 in Results for details on how the data is presented and interpreted in the manuscript.

We agree that connecting a smooth line through the sometimes-sparse data can create artefacts, but still do argue for an increase in MAP near the end of the C10 interval as the data here are robust. For the next draft submission, we will remove the smooth fit

line for intervals of sparse data (i.e., 20-50 m and 80-100 m composite depths).

RC2: 7) You are reporting false precision in $\delta$18Omw and temperature (and raw data tables too)- edit this to reflect precision within reported uncertainties.

Author Response: We report 1sd derived from ranges in sample measurements for our isotope values. As is, it is difficult to make the suggested edits without specific recommendations from the reviewer such as those included in the additional error propagation critiques below. Because the reviewer suggests using 2sd ($2\sigma$) for temperature uncertainty below (comment 11), we will now report temperature uncertainty in terms of 2se and $2\sigma$ in data tables and figures in the next draft submission. Reported precision in $\delta$18Omw will be computed using $2\sigma$ temperature uncertainty. Additionally, we plan to include all of our clumped isotope output data for equilibrated gases, heated gases, standards and samples as a supplemental file in the next draft submission.

RC2: 8) Your highest pCO2 values come from samples outside the "accepted" $\Delta$13C range for this proxy (e.g., Cotton and Sheldon, 2012). As a result, perhaps all of your es- timated values suggest low pCO2 for this period (<500ppm)? If so, does this mean C10 is non-unique, and that there is no reason to expect a shift in Hadley circulation during the mid-K? Also, why are you reporting partial uncertainties for pCO2 estimates instead of using error propagation for each component measurement (e.g., Retallack, 2009)?

Author Response: Not all of our highest pCO2 values come from samples outside "acceptable" $\Delta$13C values following Cotton and Sheldon, 2012 (e.g., sample 3-021 which suggests >500 ppmv pCO2 prior to the C1 interval). Therefore, even if these two samples in question are removed, we still clearly observe a decline in pCO2 going into the C10 interval.

We do acknowledge that any samples utilized which fall outside of the range of $\Delta$13C should be clearly marked as such. Following this, we plan on adjusting Figure 5 pCO2 symbols to reflect which samples lie outside this cutoff.

We decided to report partial uncertainties in pCO2 to give the reader a clearer idea of possible range in pCO2 error under two different sets of assumptions. For example, many of the calibrations which are applied to our data obtain quantitative paleoclimate parameters do not include calibration uncertainty and so this uncertainty cannot be propagated (e.g., Cotton and Sheldon, 2012). We propagate sampling and analytical error in our isotopic measurements (1sd). We then include two estimates of pCO2, one using MAP-derived S(z) values, and another under a broader range of all possible S(z) values without including assumptions regarding MAP-derived S(z) values. As S(z) values can have large impacts on resulting pCO2 records, we opt to include both approaches to illustrate the impact of the S(z) estimation strategy on our pCO2 record (i.e., sensitivity test).

We recognize that our reported pCO2 uncertainties could be improved by propagating uncertainties derived from individual components in quadrature to get a combined uncertainty (i.e., compute the square root of the sum of individual uncertainties as in Retallack (2009) as the reviewer suggests). In the next draft submission, we will include this approach to estimating uncertainty in pCO2 for each of our approaches (i.e., both for MAP-derived S(z) and large S(z) range approaches outlined above).

RC2: 9) How do your reported $\delta$18Omw values show changes in hydrologic cycling during the Aptian/Albian? The relatively limited isotopic range (+/-2 per mil) matches the range reported from modern environments in the same region (c.f., Zhangye and Lanzhou), and MAP shows no clear trends through time (as well as a limited range of 600-1000 mm/yr). I don't see strong evidence for either changing MAP or $\delta$18Omw across this interval (or a drop in pCO2) that would suggest a shift in Hadley Cell circulation. Are there other sites in the region to which you could compare (and perhaps make a spatial argument for the existence/location of cell boundaries; e.g., Hasegawa et al., 2012)?

Author Response: The reviewer makes a good point that while climate and hydrologic cycle variations in the C10 interval are consistent with lower temperature, atmospheric

pCO2 and perhaps shifts in the hydrologic cycle, the trends in our reconstructions tend to be washed-out, though not entirely, by uncertainty. At the very least, our study broadly captures the paleoenvironmental conditions in NW China during the Aptian-Albian, providing an important observation even without considering shorter-term variations.

We argue that higher-resolution shifts in our reconstructions likely capture an average (or seasonally consistent as discussed in the manuscript) proxy value and thus shifts cannot be appropriately compared to a modern seasonal range. While the shifts are subtle compared with uncertainty, they are consistent with carbon cycle and temperature variations for the interval and worth noting. However, we acknowledge that data from one locality is insufficient for interpreting shifts in global atmospheric circulation. Following this, in the next draft submission, we plan on toning down any language which strongly promotes the hypothesis that our records indicate shifts in Hadley Cell circulation to, for example, "may suggest" or "consistent with," etc.

Thank you for the suggestion to compare with other sites' data to further the argument for Hadley Cell shifts. Unfortunately, higher temporal resolution terrestrial temperature and $\delta$18Omw water like the records published here, does not exist for the region and narrow time interval reported here. These records are the first of their kind for mid-Cretaceous Asia.

RC2: 10) What does Figure 1 show? The placement of your sampling sites relative to one another is inconsequential to this work. Could this figure be used more effectively to show relationships between White Pagoda and other studied sites in the region (e.g., for comparison in an evaluation of Hadley extent, as above)?

Author Response: Thank you for pointing this out. We appreciate the suggestion to update this figure to better show White Pagoda in relation to other studies sites of the region such as those included in Zheng et al. (2021). This will help to better place the site within the regional bio- and chrono-stratigraphic framework. We plan on updating

this figure accordingly prior to the next draft submission.

RC2: 11) Something to consider, though maybe impractical for this work, is that most recent clumped isotope work suggests that <5 replicates is probably insufficient for appro- priately constraining $\Delta47$, and that $2\sigma$ are probably more realistic for compounded uncertainties in paleotemperature estimates (e.g., Fernandez et al., 2017; Bernasconi et al., 2021).

Author Response:Thank you for the considerations. At this stage it would be impractical to return to the lab to measure more clumped isotope values on sample material. We note that 4 replicates were measured on nearly all samples (3 replicates for sample 3A-097 only). We plan on including $2\sigma$ compound uncertainties in paleotemperature estimates (text, tables, and figures) in the next draft submission as the reviewer suggests here. Additionally, in our supplemental data tables we will include the following data columns for completeness: $\Delta47$ mean, $\Delta47$ $1\sigma$, $\Delta47$ SE, $\Delta47$ 2SE, Temperature mean, Temperature $1\sigma$, Temperature $2\sigma$, and Temperature 2SE.

Further, we recognize the recent and upcoming work in clumped isotope temperature calibration and correction, and plan to incorporate these approaches as necessary in the next draft submission, including approaches in propagating uncertainty.

---

## Author Response (AR1)

**General Response to Reviewer Commentary**

First, the authors of the manuscript entitled "Aptian-Albian clumped isotopes from northwest China: Cool temperatures, variable atmospheric $p$CO$_2$ and regional shifts in hydrologic cycle" would like to thank the three reviewers for providing focused critical evaluations of our work. Below, we directly address the reviewer's comments and include the original reviewer commentary. *Original reviewer comments are italicized below.* We largely following the advice and comments of reviewers and reference changes to the manuscript where applicable.

**Response to Review #1 (Anonymous)**

*The manuscript entitled as "Aptian-Albian clumped isotopes from northwest China: Cool temperatures, variable atmospheric pCO2 and regional shifts in hydrologic cy- cle" by Dustin T. Harper et al., present new results of pedogenic carbonate stable isotopes (δ13C, δ18O and Δ47) from the lower mid-Cretaceous (Aptian-Albian) Xiagou and Zhonggou formations, Yujingzi Basin of NW China. Authors estimate the MAT using Δ47 of carbon ad oxygen isotopic values and the MAP using CALMAG of mu- drocks, further calculate the pCO2 using MAP and δ13C of pedogenic carbonates with other parameters, and discuss the carbon and hydrologic cycles for the interval of the Aptian-Albian. This work is a new progress of the land quantitive paleoclimate in North China, even in East Asia. It would provide clues and references for the climatic re- construction of the greenhouse Cretaceous period. However, more data and evidence need to further enhance and refine, and some issues have to solve.*

*Geological data to complement. Though figure 1 shows the locations of samples in outcrop, it does not have any geological significances. It does not exhibit any strati- graphic sequences with sample horizons. In my opinion, it is important that sampling locations are plotted in a geological sketch. And it is advised that a sampling log is added.*

Much of the geologic data and sampling information for the study site was initially described in Suarez et al. (2018). We have also previously included supplemental tables in the present study detailing sample lithology and samples identified as paleosol B-horizons. However, to address the reviewer's concern for lack of stratigraphic and geologic data, we opt to include an additional supplemental figure (Fig. S1) in the resubmitted manuscript which includes lithostratigraphic columns with representative sampling locations, and images of sampled carbonate nodules, nodule features and sedimentary structures interpreted to represent pedogenesis (slickensides, root traces, ped structure etc.). Lastly, within the revised manuscript, we've opted to change figure 1 to include two outcrop photos that are representative of the lithology and *in situ* sedimentary structures indicative of well-developed paleosols.

*Data age. Authors used the organic stable carbon isotope chemostratigraphic records for the site (Suarez et al., 2018) to have the studied strata age-controlled. Albeit more than 400 organic stable carbon isotopes were correlated and suggested the Aptian-Albian in Suarez et al. (2018),*

*it is not assured and persuaded due to lack of precise age reference-point and age-index fossils. This is a common issue and prob- lem of the land materials for paleoclimatic analysis. It is cautious to make the precise correlation for the terrestrial strata and samples.*

While we agree chemostratigraphic correlation can be complicated by local influences over global variations in carbon cycle, we argue that Suarez et al. (2018) provide convincing evidence that our study sections span the "C10" carbon isotope high after Menegatti et al. (1998) and Bralower et al. (1999). Variations in $\delta^{13}C$ of organic C from the same sample collection utilized in this study have been correlated to globally representative records of carbon cycle variations.

Further, this is not the only age constraint for the Xinminpu Group. Samples from localities to the South of the Yujingzi basin have been identified as Xinminpu Group and measured for radiometric dates. These dates range from 123 ±2.6 Ma to 113.7 ±1.8 Ma (Kuang et al., 2013; Li et al., 2013).

Additionally, a recent study (Zheng et al., 2021) establishes regional ages through bio- and chronostratigraphy. This study reviews available age controls for the Lower Cretaceous in NW China and places the organic carbon isotope records of Suarez et al. (2018) (i.e., our study sections) within the chronostratigraphic framework of the region. Now that this paper has been published, we will include more details of the biostratigraphy and chronostratigraphy which better establish the age control of our study site in the next draft submission (lines 94 to 97 of tracked changes draft).

*Field description of paleosols. It is key to make sure the observed horizons are real paleosols, i.e. authors claimed the paleosols a kind of vertisol. I do not see the details of the paleosols even though authors have done analyses of CL and microfacies for the calcareous nodules. Shape, size, content, and occurrence of nodules can provide us reference for paleosols. Color, structures, and ped types can give us some information about the paleosols. Other more, vertisols are a kind of paleosols we do not easily observe and distinguish in practice in field recognition. Detailed notation of evidence seems necessary for the classification of the vertisol.*

Much of this information was established in Suarez et al. (2018) for the study sections, and for this study we avoided redundancies with respect to detailed outcrop and paleosol descriptions. However, we understand the importance of establishing the sampled paleosols as vertisols to the interpretation of our results. Following this reviewer's comment, the authors have decided to include an additional supplemental figure (also described above; Fig. S1) which includes strat columns with descriptive observations of pedogenic structures and images of hand samples. Additionally, we updated our site location (Fig. 1) based on this review and review #2. This site map shows the locality within regional geography and includes select outcrop images which show, for example, mukkara cracks, color (e.g., Retallack, 1997) and other evidence supporting our classification.

*Drilling samples for clumped isotopes. It is a good job for the clumped isotope. But it is also a problem to take powder samples from the calcretes. This is because we only need <0.1 mg for the common C-O analyses of carbonates, but we have to take over 5mg for clumped isotope analysis. It is difficult to take so much from a calcretes sample according to much experience. So, how to get the enough quantity of the powder sample may need to expain.*

Our pedogenic carbonate nodules were cm to multi-cm in scale, providing ample carbonate material for sampling. This is now stated in the revised draft (lines 165 to 166 in the tracked changes draft). The reviewer makes a good point in that it is important that the clumped isotope material that is sampled should maintain uniform $\delta^{13}C$ and $\delta^{18}O$ values. To ensure this, we measured multiple "spot" samples for $\delta^{13}C$ and $\delta^{18}O$ for each nodule, and only sampled material for clumped isotopes from areas with uniform $\delta^{13}C$ and $\delta^{18}O$ in a given sample. The average $\delta^{13}C$ and $\delta^{18}O$ values were then compared with the values obtained during the clumped isotope measurement as an indication of successful sampling. We argue that this strategy has been clearly outlined in the manuscript. In addition, the new Fig. 1 and supplementalry figure (Fig. S1) show the sizes of typical carbonate nodules sampled so that readers can have a better sense for the material sampled.

*Low temperature and low pCO2 in the Aptian-Albian. As we know from lots pa- leoceanographic and paleoclimatic achievements, the mid-Cretaceous is the hottest interval in the Phanerozoic. So, the conclusion from the authors that a low tempera- ture had been in the Aptian-Albian may need to further examine except for the short "cold snap". It may be paradox that low temperature is consistent with low pCO2 in the Aptian-Albian in Northwest China. This is because the pCO2 is almost global in nature, but the temperature in a basin, North China, was probably local record to great degree on land. Actually, we know the Cretaceous climate was not homogeneous in China.*

The coauthors humbly disagree that the scientific community has concluded that the mid-Cretaceous was the hottest interval of the Phanerozoic. Taken as a whole, yes, the Cretaceous Period was one of the warmest intervals, but this paper and many others investigate a more detailed climate record. At the stage level, many would argue that the hottest interval occurred during the Late Cretaceous near the Cenomanian-Turonian boundary (Bice et al., 2006; Hay et al., 2017). Certainly, we know that the Cretaceous was generally a warm interval (Hay et al., 2017), but a large number of publications which examine shorter-term climate variations indicate relative cool conditions at the Aptian-Albian, but also other stages such as the Valanginian (see for example, Mutterlose et al., 2009; Rodriguez-Lopez 2016; Bottini et al., 2015; Heimhofer et al., 2008; Millan et al., 2014; Vickers et al., 2019). Undoubtedly, there were large variations in the carbon cycle during this time evidenced by ocean anoxic events and variable $\delta^{13}C$ in global archives, suggesting these variations were global in nature (Menegatti et al., 1998; Bralower et al., 1999). Indeed, previous work on mid-Cretaceous climate suggests relatively cool temperatures over a large swath of latitude in Asia (Amiot et al., 2011).

We agree that surface temperatures on Earth are not homogeneous, including over a large land mass such as Asia. Also, we agree that our temperature record reflects a local signal.

Atmospheric $p$CO$_2$ is certainly a global signal when we compare atmospheric CO$_2$ mixing times with the temporal resolution of our record. However, shifts in $p$CO$_2$ are clearly linked with shifts in land surface temperatures over many intervals (e.g., Bice et al., 2006; Pagani et al., 2005; Hay et al., 2017). In the manuscript we have discussed the local versus global nature of our records, both in terms of topography and with respect to latitude, fairly extensively in the discussion section 4.3 "Latitudinal gradients of temperature and meteoric water $\delta^{18}$O for the Aptian-Albian". In this section, our temperature record was placed within a global context using latitudinal temperature profile figures (Figure 7).

Indeed, one large take-away from this work is that Cretaceous climate was not homogenous neither spatially (see discussion and figures on latitudinal temperature distribution) nor temporally. Variations were clearly occurring in terms of climate and the carbon cycle, evidenced by:
1. Shifts in global records of carbonate $\delta^{13}$C (Menegatti et al. 1998; Bralower et al. 1999; Suarez et al., 2018)
2. Multi-million year records of variable atmospheric $p$CO2 including ours and those of Bice et al. (2006) and Wang et al. (2014).
3. Mid-Cretaceous records of temperature (terrestrial and marine; for example, Bice et al., 2006; Amiot et al., 2011; O'Brien et al., 2017; this study)

*Some references are not regularly lined in text. For examples, references cited in Lines 55, 269, etc. are neither listed in the sequence of publishing time nor of surname letter.*

Thank you for pointing this out. We addressed this in our resubmission.

**Response to Review #2 (Anonymous)**

*Harper et al. use pedogenic carbonates and paleosol elemental geochemistry to de- velop new temperature, precipitation, and pCO2 estimates of "mid" Cretaceous pale- oclimate in northwest China. They use these records to confirm previous reconstruc- tions, and to suggest that these conditions may represent examples of thresholds in shifting Hadley circulation at this time.*

*Overall this work contributes useful new data for the region and time period, but could benefit from a refocus of the work within geographic context and with additional dis- cussion of regional climate and potential uncertainties. Below are some comments on particular aspects of the work that could be improved or reevaluated before publication. Comments and revisions:*

*1) Improve editing of the manuscript throughout (incorrect agreement, missing words, etc.).*

Thank you for pointing this out. We addressed these errors in our resubmission.

*2) Is the -8.23 per mil correction for δ13Ca reasonable for this period, given the ex- istence of glendonites and the low temperatures and low pCO2? Cooler, low-pCO2 periods during the Cenozoic have substantially higher δ13Ca values (-5.5 to -6.5 per mil; Tipple et al., 2010), which may change your eventual pCO2 estimates (make them slightly higher?).*

For our study, a −8.23 ‰ correction yields $\delta^{13}C_a$ values which range from −5.38 ‰ to −4.18 ‰ (Table 4). These values are indeed higher than those the reviewer lists from Tipple et al., 2010). Perhaps the reviewer was thinking that −8.23 ‰ was the applied $\delta^{13}C_a$ value for all sample calculations? We can assure you this is not the case. We now more clearly lay out the applied offset and resulting $\delta^{13}C_a$ values in the resubmission (lines 253-254 in tracked revision draft).

If the reviewer was intending on recommending using lower values in the range of the cool Cenozoic values established by Tipple et al. (2010) (i.e., −6.5‰ to −5.5 ‰), we argue that given the relative differences in climate between the cool intervals of the Cenozoic and the cool intervals of the Cretaceous (i.e., the cool Cretaceous was likely warmer than cool Cenozoic; Hay et al. 2017; Bice et al., 2006; Westerhold et al., 2020), our slightly higher (~+1.0‰) $\delta^{13}C_a$ values are appropriate. If, however, lower values (~−6.0 ‰) were applied, the reviewer is correct in stating that $pCO_2$ estimates would tend to increase marginally.

As an example, if the $\delta^{13}C_a$ value for sample 4-038 was adjusted to −6.0 ‰ (mid-point in the range suggested by the reviewer), the reconstructed atmospheric $pCO_2$ value would shift from 682 to 712 ppmv.

*3) CALMAG is an elemental ratio, and should not be reported in % (e.g., Nordt et al., 2010).*

Thank you for pointing this out. We removed "%" in our resubmitted manuscript.

*4) These cathodoluminescence images are concerning. High luminescence indicates substantial Mn, Fe, etc. which is usually indicative of diagenesis (e.g., Driese & Mora, 1993; Budd et al., 2002), which appears to be what you sampled. Also, the final image (Figure 3, sample 6-042) is incorrectly illuminated and the bright region is just showing an incident beam from the CL (which is not calibrated across the surface). You may want to reevaluate your data to distinguish between samples selected from different regions of the carbonate nodules, and confirm that the presented data are from primary materials.*

As the third reviewer suggests, these CL images were taken with conditions for high luminescence sensitivity (e.g., He chamber). Pedogenic carbonate nodules likely formed over many seasons and as these paleosols are interpreted as vertisol, undoubtedly, the water table likely fluctuated seasonally resulting in calcite precipitation and stabilization both during periods of saturation (below the water table) and in the vadose zone. High luminescence does not always indicate degree of diagenesis evidenced by the non-luminescent clear fracture-filled spar in sample 3-021 (see Fig. 3 of resubmitted manuscript). This spar tended to have an isotopic diagenetic signature (lower $\delta^{18}O$ and $\delta^{13}C$ values) when compared to homogenous

isotope values of the highly-luminescent, but homogenous, nodule carbonate (Figure 4). In addition, the luminescence may be expected in these soils if they were seasonally saturated, and may not indicate of degree of diagenesis. This has now been articulated in lines 148 to 151 of the tracked changes draft). The Budd et al. (2002) reference that Reviewer 2 recommends describes variable luminescence in addition to discernible disequilibrium between $\delta^{13}C_{carb}$ and $\delta^{13}C_{org}$ as evidence for diagenetic alteration from environmental values. Interestingly, even samples that show higher degree of heterogeneity in luminescence (sample 06-042), show fairly homogenous $\delta^{13}C$ and $\delta^{18}O$, with $\Delta^{13}C$ values that do not indicate disequilibrium. Samples that do show some degree of disequilibrium in carbon isotope values were removed from use for calculation of $pCO_2$. However, we are comfortable including the clumped isotope temperature value because clumped isotope derived temperatures are independent of stable isotope values, so even if there was some amount of early diagenesis, it likely still represents near surface temperatures (i.e., pre-burial) as pedogenic carbonates tend to form over thousands of years (Giles et al., 1966).

We acknowledge that sample 6-042 is poorly illuminated and shows the incident beam. To address the sub-optimal quality of CL imaging in the manuscript, we now include new images in the revised submission. These new images reveal dull to moderate luminescent carbonate matrix and carbonate nodules with displacive luminescent fractures. These fractures were avoided when sampling for isotope values used in paleoenvironmental proxies.

We do indeed evaluate our data by distinguishing $\delta^{13}C$ and $\delta^{18}O$ from different regions of the nodules (micritic nodule vs. spar) in Figure 4 as the reviewer suggests. Generally, apparent secondary calcite phases (e.g., spar) are offset from the ranges in multi-spot stable isotope values of micritic, likely primary, phases of calcite (Figure 4).

*5) Why do you need Figure 6 showing different paleogeographies? Unless you add in simulations of MAT and δ18Omw as an overlay (e.g., Zhou et al., 2008; Hasagawa et al., 2012), this doesn't really contribute to the paper. Instead just rely on Figure 7 to show what you're arguing with respect to paleogeography, and perhaps expand the discussion of this point to match.*

We respectfully lobby for this figure to remain. We argue this is an important figure for the manuscript and intend to keep it in the next draft submission for the following reasons:
1) Allows readers to place the study location within a greater tectonic framework of the time. Including the paleogeography of the study can help readers more clearly understand paleoenvironmental setting and potential complication with regards to our interpretation of paleoenvironment. For example, it provides a scale for paleolocation uncertainty.
2) This figure provides a visual aid of ranges of paleolatitude for key sites in Asia for the Aptian-Albian which has not been previously done. Much of the literature which describes Aptian-Albian climate in Asia relies on these reconstructions. These studies may tend to exaggerate cool or warm conditions for a region accordingly if they do not consider all possible reconstructions.

3) A map view of the possible reconstructions helps with visualizing the potential geography that may impact stable isotope and paleotemperatures. For example, a more inland reconstruction may suggest greater continentality influence in temperature and isotopic composition of precipitation.

*6) CALMAG values reported are different in the Results vs. the Discussion- does the version in the Discussion and in Figure 5 include non-B horizons? Check this and revise (or specify) as needed. The CALMAG-derived MAP fit in Figure 5 is also overly smoothed- there are not enough data points for the level of smoothing (moving average I assume?), which results in data artefacts like the curve at ~40m.*

In the results and Supplemental Table S4 we describe and list CALMAG values for all available samples. In Table 3 and Figure 5 (as well as in the discussion), we only include paleosol B-horizon samples which are within the range appropriate to apply MAP-calibration following recommendations by Nordt and Driess (2010). Please see section 3.3 in Results for details on how the data is presented and interpreted in the manuscript.

We agree that connecting a smooth line through the sometimes-sparse data can create artefacts, but still do argue for an increase in MAP near the end of the C10 interval as the data here are robust. For the next draft submission, we will remove the smooth fit line for intervals of sparse data (i.e., 20-50 m and 80-100 m composite depths). The smoothing is indeed a 3 point moving average.

*7) You are reporting false precision in δ18Omw and temperature (and raw data tables too)- edit this to reflect precision within reported uncertainties.*

We previously reported 1sd derived from ranges in sample measurements for our isotope values. As is, it is difficult to make the suggested edits without specific recommendations from the reviewer such as those included in the additional error propagation critiques below. Because the reviewer suggests using 2sd ($2\sigma$) for temperature uncertainty below (comment #11), we now report temperature and $\delta^{18}O_{mw}$ uncertainty in terms of $2\sigma$ in data tables and figures. Reported precision in $\delta^{18}O_{mw}$ is computed using $2\sigma$ temperature uncertainty. Additionally, we include all of our clumped isotope output data for gases, standards and samples as a supplemental table (Table S2).

*8) Your highest pCO2 values come from samples outside the "accepted" Δ13C range for this proxy (e.g., Cotton and Sheldon, 2012). As a result, perhaps all of your es- timated values suggest low pCO2 for this period (<500ppm)? If so, does this mean C10 is non-unique, and that there is no reason to expect a shift in Hadley circulation during the mid-K? Also, why are you reporting partial uncertainties for pCO2 estimates instead of using error propagation for each component measurement (e.g., Retallack, 2009)?*

Not all of our highest $pCO_2$ values come from samples outside "acceptable" $\Delta^{13}C$ values following Cotton and Sheldon, 2012 (e.g., sample 3-021 which suggests >500 ppmv $pCO_2$ prior to the C1 interval). Therefore, even if these two samples in question are removed, we still clearly observe a decline in $pCO_2$ going into the C10 interval.

We do acknowledge that any samples utilized which fall outside of the range of $\Delta^{13}C$ should be clearly marked as such. Following this, we adjusted Figure 5 $pCO_2$ symbols to reflect which samples lie just outside this cutoff.

We previously decided to report partial uncertainties in $pCO_2$ to give the reader a clearer idea of possible range in $pCO_2$ error under two different sets of assumptions. For example, many of the calibrations which are applied to our data obtain quantitative paleoclimate parameters do not include calibration uncertainty and so this uncertainty cannot be propagated (e.g., Cotton and Sheldon, 2012).

To test the sensitivity of our $pCO_2$ record to S(z), we provided two reconstructions: one using MAP-derived S(z) values, and another under a broader range of all possible S(z) values without including assumptions regarding MAP-derived S(z) values. As S(z) values can have large impacts on resulting $pCO_2$ records, we opt to include both approaches to illustrate the impact of the S(z) estimation strategy on our $pCO_2$ record (i.e., sensitivity test). Our approach is now more clearly laid out in lines 453-464 of the tracked changes document.

We recognize that our reported $pCO_2$ uncertainties could be improved by propagating uncertainties derived from individual components in quadrature to get a combined uncertainty (i.e., compute the square root of the sum of individual uncertainties as in Retallack (2009) as the reviewer suggests). This is now included in the text (see section 4.4), tables and figures.

*9) How do your reported δ18Omw values show changes in hydrologic cycling during the Aptian/Albian? The relatively limited isotopic range (+/-2 per mil) matches the range reported from modern environments in the same region (c.f., Zhangye and Lanzhou), and MAP shows no clear trends through time (as well as a limited range of 600-1000 mm/yr). I don't see strong evidence for either changing MAP or δ18Omw across this interval (or a drop in pCO2) that would suggest a shift in Hadley Cell circulation. Are there other sites in the region to which you could compare (and perhaps make a spatial argument for the existence/location of cell boundaries; e.g., Hasegawa et al., 2012)?*

The reviewer makes a good point that while climate and hydrologic cycle variations in the C10 interval are consistent with lower temperature, atmospheric $pCO_2$ and perhaps shifts in the hydrologic cycle, the trends in our reconstructions tend to be washed-out, though not entirely, by uncertainty. At the very least, our study broadly captures the paleoenvironmental conditions in NW China during the Aptian-Albian, providing an important observation even without considering shorter-term variations.

We argue that higher-resolution shifts in our reconstructions likely capture an average (or seasonally consistent as discussed in the manuscript) proxy value and thus shifts cannot be appropriately compared to a modern seasonal range. A sentence has been added to articulate this point (lines 506 to 508 in tracked changes draft). While the shifts are subtle compared with uncertainty, they are consistent with carbon cycle and temperature variations for the interval and worth noting. However, we acknowledge that data from one locality is insufficient for interpreting shifts in global atmospheric circulation. Following this, in the resubmission, we toned down language which strongly promotes the hypothesis that our records indicate shifts in Hadley Cell circulation to, for example, "suggest subtle" (line 505 of tracked changes draft). We have also added a note about relative magnitude of $p$CO$_2$ change compared with reconstruction error to the discussion (see lines 544 to 547 in the tracked changes draft).

Thank you for the suggestion to compare with other sites' data to further the argument for Hadley Cell shifts. Unfortunately, higher temporal resolution terrestrial temperature and $\delta^{18}$O$_{mw}$ water like the records published here, does not exist for the region and narrow time interval reported here. These records are the first of their kind for mid-Cretaceous Asia, and the reviewer's comments highlight the need for greater spatial and temporal climate data from the continental interior of Asia during this time period.

*10) What does Figure 1 show? The placement of your sampling sites relative to one another is inconsequential to this work. Could this figure be used more effectively to show relationships between White Pagoda and other studied sites in the region (e.g., for comparison in an evaluation of Hadley extent, as above)?*

Following this reviewer's comment and reviewer 1's suggestions, we have decided to change Fig. 1 by: 1) removing the placement of sampling sites in map view, 2) including a new map which better illustrates the region for comparison to other localities and neighboring provinces, and 3) showing outcrop images which include paleosol features. However, we still show the locality in terms of regional geography.

*11) Something to consider, though maybe impractical for this work, is that most recent clumped isotope work suggests that <5 replicates is probably insufficient for appro- priately constraining Δ47, and that 2σ are probably more realistic for compounded uncertainties in paleotemperature estimates (e.g., Fernandez et al., 2017; Bernasconi et al., 2021).*

Thank you for the considerations. At this stage it would be impractical to return to the lab to measure more clumped isotope values on sample material, particularly due to travel and lab access restrictions. We note that 4 replicates were measured on nearly all samples (3 replicates for sample 3A-097 only). We now include 2σ uncertainties in paleotemperature estimates (text, tables, and figures) as the reviewer suggests here. Additionally, we include $\Delta_{47}$ 2SE in our text, tables and figures. This information has been added to lines 204-207 in the tracked changes draft.

**Response to Review #3 (Hilary Corlett)**

*I was asked to provide a review focused on the reviewer comment below and Figure 3 from cp-2020-152.*

*The reviewer comment is: These cathodoluminescence images are concerning. High luminescence indicates substantial Mn, Fe, etc. which is usually indicative of diage- nesis (e.g., Driese & Mora, 1993; Budd et al., 2002), which appears to be what you sampled. Also, the final image (Figure 3, sample 6-042) is incorrectly illuminated and the bright region is just showing an incident beam from the CL (which is not calibrated across the surface). You may want to reevaluate your data to distinguish between samples selected from different regions of the carbonate nodules, and confirm that the presented data are from primary materials.*

*Please find my review below:*

*I would agree that the images are concerning. They looked odd to me and when I read the text, the method of "Macroscale imaging through the 50 mm top window of the chamber was carried out using a 16 Mpx Canon EOS SL1 DSLR camera with a macro lens suspended over the CL chamber" may explain why the luminescence is not what you expect from a standard CL image. These images would normally be taken with a C-mounted microscope camera or C-mount DSLR, or if C-mount is missing, then you would use an ocular mounted USB microscope camera or DSLR. The luminescence in the photos may be exaggerated in some way, which may be why the reviewer is concerned. Also, the review is correct about the last image where it is showing the incident beam. This image should not be used.*

*The cause of luminescence is Mn, and Fe is more of a quenching element. If these nodules formed under slightly reduced conditions, you would expect there to be some uniform, dull orange luminescence. In a pedogenic environment, you may expect there to be some luminescence because pore waters forming these nodules may be relatively Fe poor, if the Fe is oxidized, as is the case in these types of environments. I wouldn't agree (with the reviewer's comment) that any luminescence at all means diagenesis. Given the fact that the zones indicated as "primary" have a uniform luminescence in the first two pictures, as long as the isotope data is fairly consistent (i.e. showing several data points in a narrow range), I wouldn't say these have been diagenetically overprinted. The third image has a more mottled appearance and therefore, may have experienced some diagenesis. It is impossible to say anything about the fourth image because it is just showing the incident beam and causing one area to be more brightly illuminated than the surrounding. The best example here is the first image with a uniform orange/yellow luminescence of the nodule, marked as primary, and the fracture with mottled appearance marked as secondary.*

*The final word from me would be: 1) get rid of the last image, 2) mention that the method of photography used here may have resulted in overexposure of the CL images, and 3) only images 1 and 2 are convincing as "primary" formed from slightly reduced water, enriched in Mn. If the*

*third nodule also resulted in isotope data that falls within the range of the first two, then this mottled appearance may just indicate this nodule has experienced minor diagenetic alteration or when it formed, it incorporated some of the matrix into the nodule, giving it a less uniform luminescence.*

*For context, I am a carbonate sedimentologist with a primary research focus on dia- genesis. I am familiar with CL of carbonates but have not read the rest of the paper as I was only asked to provide this limited review. The review process required me to fill in the recommendations, but please note that my recommendation is limited to the CL imagery and interpretation. For all of my other recommendations, I defaulted to the median.*

*Hilary Corlett*

We greatly appreciate the comments on our CL work and on the previous review of our CL work. The specific recommendations are clear and make sense. They have been implemented in the resubmission in the following ways:

1) We removed the last image and have taken new images with a C-mounted Olympus microscope camera (DP73) as Dr. Corlett suggests. See lines 133-145 in tracked changes draft for updated approach.
2) All of the previous CL macroscopic images have been removed from Figure 3
3) We appreciate and agree with Dr. Corlett's interpretation of "primary" luminescent material from Mn-enriched reduced water sources, and that the mottled appearance of nodule 3 (sample 3E-001) compared with samples 3-021 and 3A-097 is due to differences in matrix incorporation. This is bolstered by our isotope data for 3E-001 which falls within ranges of samples 3-021 and 3E-001, with comparable $\delta^{13}C$ and $\delta^{18}O$ standard deviations among multi-spot samples. We updated the text to include these arguments and interpretations on lines 189-190 and in section 3.1 of the tracked changes draft.

**References Cited**

Amiot, R., Wang, X., Zhou, Z., Wang, X., Buffetaut, E., Lecuyer, C., Ding, Z., Fluteau, F., Hibino, T., Kusuhashi, N., Mo, J., Suteethorn, V., Wang, Y., Xu, X., and Zhang, F.: Oxygen isotopes of East Asian dinosaurs reveal exceptionally cold Early Cretaceous climates, Proc Natl Acad Sci U S A, 108, 5179-5183, 10.1073/pnas.1011369108, 2011.

Bice, K. L., Birgel, D., Meyers, P. A., Dahl, K. A., Hinrichs, K. U., & Norris, R. D. (2006). A multiple proxy and model study of Cretaceous upper ocean temperatures and atmospheric CO2 concentrations. *Paleoceanography*, *21*(2).

Bottini, C., Erba, E., Tiraboschi, D., Jenkyns, H. C., Schouten, S., & Sinninghe Damsté, J. S. (2015). Climate variability and ocean fertility during the Aptian Stage. *Climate of the Past*, *11*(3), 383-402.

Bralower, T. J., CoBabe, E., Clement, B., Sliter, W. V., Osburn, C. L., and Longoria, J.: The record of global change in mid-Cretaceous (Barremian-Albian) sections from the Sierra Madre, Northeastern Mexico, Journal of Foraminiferal Research, 29, 20, 1999.

Budd, D. A., Pack, S. M., & Fogel, M. L. (2002). The destruction of paleoclimatic isotopic signals in Pleistocene carbonate soil nodules of Western Australia. *Palaeogeography, Palaeoclimatology, Palaeoecology*, *188*(3-4), 249-273.

Cotton, J. M., and Sheldon, N. D.: New constraints on using paleosols to reconstruct atmospheric pCO2, Geological Society of America Bulletin, 124, 1411-1423, 10.1130/b30607.1, 2012.

Gile, L. H., Peterson, F. F., & Grossman, R. B. (1966). Morphological and genetic sequences of carbonate accumulation in desert soils. *Soil Science*, *101*(5), 347-360.

Hay, W. W. (2017). Toward understanding Cretaceous climate—An updated review. *Science China Earth Sciences*, *60*(1), 5-19.

Heimhofer, U., Adatte, T., Hochuli, P. A., Burla, S., & Weissert, H. (2008). Coastal sediments from the Algarve: low-latitude climate archive for the Aptian-Albian. *International Journal of Earth Sciences*, *97*(4), 785-797.

Kuang, H. W., Liu, Y. Q., Liu, Y. X., Peng, N., Xu, H., and Dong, C.: Stratigraphy and depositional palaeogeography of the Early Cretaceous basins in Da Hinggan Mountains–Mongolia orogenic belt and its neighboring areas, Geological Bulletin of China 32, 22, 2013.Li et al., 2013

Menegatti, A. P., Weissert, H., Brown, R. S., Tyson, R. V., Farrimond, P., Strasser, A., and Caron, M.: High-resolution δ13C stratigraphy through the Early Aptian "Livello selli" of the Alpine tethys, Paleoceanography, 13, 530-545, 10.1029/98pa01793, 1998.

Millán, M. I., Weissert, H. J., & López-Horgue, M. A. (2014). Expression of the Late Aptian cold snaps and the OAE1b in a highly subsiding carbonate platform (Aralar, northern Spain). *Palaeogeography, Palaeoclimatology, Palaeoecology*, *411*, 167-179.

Mutterlose, J., Bornemann, A., and Herrle, J.: The Aptian – Albian cold snap: Evidence for "mid" Cretaceous icehouse interludes, Neues Jahrbuch für Geologie und Paläontologie - Abhandlungen, 252, 217-225, 10.1127/0077-7749/2009/0252-0217,

Nordt, L. C., and Driese, S. D.: New weathering index improves paleorainfall estimates from Vertisols, Geology, 38, 407-410, 10.1130/g30689.1, 2010.

O'Brien, C. L., Robinson, S. A., Pancost, R. D., Damste, J. S. S., Schouten, S., Lunt, D. J., ... & Wrobel, N. E. (2017). Cretaceous sea-surface temperature evolution: Constraints from TEX86 and planktonic foraminiferal oxygen isotopes. *Earth-Science Reviews*, *172*, 224-247.

Pagani, M., Zachos, J., Freeman, K., Tipple, B., and Bohaty, S.: Marked Decline in Atmospheric Carbon Dioxide Concentrations During the Paleogene, Science, 309, 2005.

Retallack, G. J.: Colour guide to paleosols, edited by: Ltd., J. W. a. S., 1997.

Retallack, G. J. (2009). Refining a pedogenic-carbonate CO2 paleobarometer to quantify a middle Miocene greenhouse spike. *Palaeogeography, Palaeoclimatology, Palaeoecology*, *281*(1-2), 57-65.

Rodríguez-López, J. P., Liesa, C. L., Pardo, G., Meléndez, N., Soria, A. R., & Skilling, I. (2016). Glacial dropstones in the western Tethys during the late Aptian–early Albian cold snap: Palaeoclimate and palaeogeographic implications for the mid-Cretaceous. *Palaeogeography, Palaeoclimatology, Palaeoecology*, *452*, 11-27.

Suarez, M. B., Milder, T., Peng, N., Suarez, C. A., You, H., Li, D., and Dodson, P.: Chemostratigraphy of the Lower Cretaceous dinosaur-bearing Xiagou and Zhonggou Formations, Yujingzi Basin, Northwest China, Journal of Vertebrate Paleontology, 38, 10.1080/02724634.2017.1510412, 2018.

Vickers, M. L., Price, G. D., Jerrett, R. M., Sutton, P., Watkinson, M. P., and FitzPatrick, M.: The duration and magnitude of Cretaceous cool events: Evidence from the northern high latitudes, GSA Bulletin, 131, 1979-1994, 10.1130/b35074.1, 2019.

Wang, Y., Huang, C., Sun, B., Quan, C., Wu, J., & Lin, Z. (2014). Paleo-CO2 variation trends and the Cretaceous greenhouse climate. *Earth-Science Reviews*, *129*, 136-147.

Westerhold, Thomas, et al. "An astronomically dated record of Earth's climate and its predictability over the last 66 million years." *Science* 369.6509 (2020): 1383-1387.

Zheng, D., Wang, H., Li, S., Wang, B., Jarzembowski, E. A., Dong, C., ... & Zhang, H. (2020). Synthesis of a chrono-and biostratigraphical framework for the Lower Cretaceous of Jiuquan, NW China: Implications for major evolutionary events. *Earth-Science Reviews*, 103474.

---

## Author Response (AR2)

Author Response to Review Submitted on 11 Jun 2021 by Anonymous Referee #2

Reviewer Comments:
The authors have done a nice job of addressing the suggested revisions from my initial review, including a better consideration of uncertainty and error reporting, substantially revised figures, and a more thorough discussion of implications. I think with some minor technical corrections (e.g., typos) and updates (e.g., precision reporting; d18Osw = -9.5 per mil, not -9.47 per mil), this work is acceptable for publication.

Author Response:
Thank you to the reviewers for their feedback which greatly improved the initial version of our manuscript. Following this final reviewer's comment, we have changed our reported precision for $\delta^{18}O_{mw}$ reconstructions in the text/tables and proofread for typos.